# WEIGHTED FAIR REGRESSION UNDER SELECTION BIAS

## ABSTRACT

Selection bias is a prevalent challenge in real-world data analysis, often stemming from biased historical censoring policies. While there is a growing body of literature on fairness in mitigating accuracy disparities, few studies have considered the potential impact of selection bias in training data. Depending on the selection mechanism, significant differences can arise between the population distribution and the training data distribution. Therefore, the training fairness metric can be heavily biased, leading to unfair learning. To address this issue under the fair regression problem, we propose weighting adjustments in the fairness constraint, which results in a novel fair regression estimator. Despite non-convexity, we derive an efficient algorithm to obtain a globally optimal solution. This work pioneers the integration of weighting adjustments into the fair regression problem, introducing a novel methodology to constrain accuracy disparities under arbitrary thresholds.

## 1 INTRODUCTION

The use of machine learning methods has become increasingly popular in critical sectors such as employment, education, healthcare, and criminal justice. However, bias and unfairness arising from prediction algorithms have led to growing concerns. Notably, accuracy disparities across different demographic groups have been observed in various scenarios (Barocas & Selbst, 2016). For example, Seyyed-Kalantari et al. (2021) showed AI systems consistently underdiagnosed historically underserved patient populations, such as female patients and black patients. In another case, analysis conducted by Julia Angwin & Kirchner (2016) highlights that the recidivism risk scale of COMPAS software exhibits a higher false positive rate for black defenders compared to their white counterparts.

While recent efforts have been devoted to mitigating algorithmic accuracy disparity (Agarwal et al., 2019; Donini et al., 2018; Chi et al., 2021), most of them presume the training data unbiasedly represents the population, which is a usually violated assumption in the real world. In practice, datasets may suffer from *selection bias*, originating from historically biased decision-making, thereby distorting their representativeness of the population. Machine learning algorithms trained on such datasets induce bias on population-level estimands, posing a remaining violation of fairness, sometimes referred to as *residual unfairness* (Kallus & Zhou, 2018). Many sectors grappling with AI fairness concerns have encountered such biases in their data collection procedures. For instance, data on loan default is typically collected only from applicants who were historically approved. This dataset is then used to develop loan approval policies applicable to all applicants, introducing an inherent bias (Kallus & Zhou, 2018).

In this paper, we introduce weighting adjustments to the model trained on a dataset under selection bias, thereby extending the model accuracy parity to the population level. Our primary focus is on regression problems with continuous outcomes ($Y$) and a binary protected attribute ($A$). Some prior works (e.g., Chi et al., 2021; Zhao, 2021; Oneto et al., 2019) have made strides in mitigating regressor accuracy disparity without considering selection bias. Chi et al. (2021) reduced the accuracy disparity of a regressor through adversarially training a fair representation that minimizes the distance between two group distributions while maximizing the representation's predictive power. Similarly, Zhao (2021) showed for Lipschitz continuous predictors, the accuracy disparity can be reduced by training the model on a fair representation that minimizes the Wasserstein distance between two group distributions. Oneto et al. (2019) converted the regression problem to binary classifications by discretizing the continuous label and adopted the same technique as in Donini et al. (2018), which regulates the disparity of true positive rate across the sensitive groups. However, research in

mitigating accuracy disparity in regression under selection bias remains relatively scarce. We present a few related works below.

**Related work**  Many studies addressing the impact of selection bias on algorithmic fairness focus on scenarios where the training data consists of samples from the population of interest $p_{\mathcal{T}}$, with their covariates $(X)$ always observed and outcomes $(Y)$ selectively observed (Kallus & Zhou, 2018; Du et al., 2022; Coston et al., 2021b). Kallus & Zhou (2018) used an estimated propensity score ratio between the target and training distribution to address the covariate shift caused by selection. Du et al. (2022) considered a special sample selection process and used the Inverse Mills Ratio derived from the Heckman Model to correct the selection bias. Coston et al. (2021b) used extrapolation to infer the unobserved labels and de-biased the fairness estimation over a Rashomon Set of the model. In practice, however, data subject to selection may also exhibit incompleteness in their covariates $(X)$. We provide a recruitment example to illustrate this scenario in Example 1 below. As far as we know, only Zhang & Long (2021) considered this more general setting of selection bias characterized by incomplete covariates $(X)$ and/or outcome $(Y)$ in fairness-related literature. They utilize inverse probability weighting (IPW) to assess the bias in fairness metric estimands while not aiming to control the algorithmic unfairness. Among all these works, the most relevant one we could find is Du et al. (2022), which addresses the selection bias issue in the regression problem in order to construct a fair regressor. However, the method in Du et al. (2022) is developed under a more restrictive setting than ours, assuming that the propensity score follows the probit model of a linear function of the covariates, which is not required in our assumption. Moreover, Du et al. (2022) formulates the constrained optimization with a Lagrangian form that is similar to ours, but controls the trade-off by manually tuning the dual variable, while we develop an algorithm that constrains the accuracy disparities under any predetermined thresholds.

**Our contributions:**  This work offers a few novel contributions. Firstly, we introduce weighting techniques to address *selection bias* in fair regression problems, a topic that has received limited attention in the literature. Our method is developed under a more general setting than previous studies in this area. Secondly, there is a lack of research on constraining accuracy disparity in regression problems, and none have considered constraining accuracy disparity below arbitrary thresholds. In this work, we propose a computationally efficient algorithm that limits training accuracy disparity below arbitrary thresholds and provides a globally optimal solution despite the non-convexity of the constrained optimization. Thirdly, we propose a novel parameter tuning method by approximating leave-one-out cross-validation in constrained regression problems, significantly reducing computation time compared to commonly adopted $K$-fold cross-validation.

## 2  PROBLEM STATEMENT

Let $(Z, A)$ be a vector of random variables, where $A$ is a protected attribute, and $Z := \{X, Y\} \in \mathcal{X} \times \mathcal{Y}$ consists of predictors $X$ and response variable $Y$. For simplicity, we assume $A$ to be binary such that $A \in \{0, 1\}$. Let $g : \mathcal{X} \to \mathcal{Y}$ denote a predictive function. We define $R_{\mathcal{T}}(g) := \mathbf{E}_{\mathcal{T}}\{\ell(g; Z)\}$, where $\ell$ is a loss function, and the expectation $\mathbf{E}_{\mathcal{T}}$ is taken over a *population distribution* $p_{\mathcal{T}}$. In this paper, we focus on the regression problem, and therefore, we let $\ell$ to be the $\mathcal{L}_2$ loss function, i.e., $\ell(g; Z) = \big(g(X) - Y\big)^2$. Given an appropriate function class $\mathcal{G}$, we define the *population fair regression function* with fairness constraint set $\mathcal{C}$ as

$$g_\star := \underset{g \in \mathcal{G} \cap \mathcal{C}}{\operatorname{argmin}} \, R_{\mathcal{T}}(g).$$

**Fairness notion**  The fairness constraint is often in the form $\Delta(g) \leq \delta$ for some discrepancy measure $\Delta : \mathcal{G} \to [0, \infty)$ and some $\delta \geq 0$. In this paper, we consider the mean squared error (MSE) disparity Chi et al. (2021); Zhang & Long (2021) as our fairness metric:

$$\Delta(g) = |\mathcal{E}_1(g) - \mathcal{E}_0(g)|, \quad \text{where} \quad \mathcal{E}_a(g) = \mathbf{E}_{\mathcal{T}}\{\ell(g; Z)|A = a\} \quad \text{for} \quad a = 0, 1,$$

is the conditional MSE. Our target is

$$g_\star = \underset{g \in \mathcal{G}}{\operatorname{argmin}} \Big\{ \mathbf{E}_{\mathcal{T}}\{\ell(g(X); Y)\} : \Delta(g) \leq \delta \Big\}. \tag{1}$$

If we are given independent and identically distributed (i.i.d.) training samples $\{(Z_i, A_i)\}_{i=1}^n$ from $p_{\mathcal{T}}$, then $\mathcal{E}_a(g)$ can be estimated consistently by the data strata with $A = a$, and therefore, a straightforward approach is to replace the risk and the expectation terms in the constraint by the direct sample analogs, i.e., (conditional) averages.

**Training data distribution under selection bias**   We assume the training samples are collected under potential selection bias. We use $S_i \in \{0, 1\}$ to indicate whether the $i$-th datum remains fully observed after selection. When $S_i = 1$, we obtain $Z_i$. Conversely, when $S_i = 0$, part of $Z_i$ is not observed. The conditional distribution of $Z_i$ given $S_i = 1$ is denoted by $p_{\mathcal{D}}$, which is also referred to as *training distribution*.

Due to the selection process, the training distribution $p_{\mathcal{D}}$ is not necessarily the same as the population distribution $p_{\mathcal{T}}$, and the empirical group risk can be biased for estimating the conditional risk. As such, adjustment will be needed to correct the selection bias and its effect on the estimated (conditional) risk. The issue has been discussed in Zhang & Long (2021); Kallus & Zhou (2018); Du et al. (2022); Coston et al. (2021a). We present an illustrative example in Example 1.

**Example 1** (Recruitment pre-screening)**.** *Consider a recruitment system where the success of a job interview ($Y$) is predicted based on a candidate's major ($X_1$), undergraduate GPA ($X_2$), and an assessment score ($X_3$), with gender as the protected attribute $A$. A pre-screening process is first applied to filter candidates based on $X_1$ and $X_2$. If a candidate fails this pre-screening ($S = 0$), their $X_3$ and $Y$ will become unrecorded (i.e., not selected). Therefore, if the pre-screening policy favors majors predominantly chosen by one gender group ($A = 1$), the other group ($A = 0$) may be underrepresented in the training data ($p_{\mathcal{D}}$). Consequently, models trained on such data can propagate biased estimates of risk $R_{\mathcal{T}}$ and fairness metrics in the population $p_{\mathcal{T}}$.*

**Standard selection mechanism**   We need the following assumption for the training sample selection mechanism.

**Assumption 1.** *There exists a sub-vector $U$ of $Z$, which is not subject to selection, such that,*

$$\mathbf{E}_{\mathcal{T}}\{\ell(g; Z)|A, U\} = \mathbf{E}_{\mathcal{D}}\{\ell(g; Z)|A, U\}, \quad \text{for all } g \in \mathcal{G}. \tag{2}$$

The condition (2) (only based on conditional expectations) is implied by the stronger condition that $p_{\mathcal{T}}(Z|U, A) = p_{\mathcal{D}}(Z|U, A)$, which is commonly assumed in many related works (Coston et al., 2021a; Zhang & Long, 2021; Kallus & Zhou, 2018). Note that this stronger condition, which states that the selection mechanism $S$ is independent of $Z$ given the always observable $U$ and the protected attribute $A$, is closely related to the missing-at-random condition when $S$ is considered as an indicator of observation. As for Example 1, Assumption 1 is satisfied as the pre-screening (selection process) depends on the candidate's major ($X_1$) and undergraduate GPA ($X_2$), which are always observable. If the selection process depends on some covariates that are not always fully observed, Assumption 1 may be violated. A numerical experiment under this scenario is presented in Section 5.

## 3   WEIGHTING ADJUSTMENTS

Note that the selection mechanism in (2) implies that

$$\mathbf{E}_{\mathcal{T}}\{\ell(g; Z)|A = a\} = \mathbf{E}_{\mathcal{T}}\{\mathbf{E}_{\mathcal{T}}\{\ell(g; Z)|U, A = a\}|A = a\} = \mathbf{E}_{\mathcal{T}}\{\mathbf{E}_{\mathcal{D}}\{\ell(g; Z)|U, A = a\}|A = a\}.$$

The outermost (conditional) expectation in the RHS is taken over $U$, which can be estimated by the data as $(U, A)$ is not subject to selection and, hence, is always observed. Let $r(g, U, a) = \mathbf{E}_{\mathcal{T}}\{\ell(g; Z)|U, A = a\} = \mathbf{E}_{\mathcal{D}}\{\ell(g; Z)|U, A = a\}$. We would like to find a weighted estimator for $\mathcal{E}_a(g)$ using the fully observed data in the form of

$$\tilde{\mathcal{E}}_a(g, W) := \frac{1}{n} \sum_{i=1}^n S_i W_i \mathbb{I}(A_i = a) \ell(g; Z_i), \tag{3}$$

with some appropriate weights $\{W_i\}$, and $\mathbb{I}(\cdot)$ denotes the indicator function. One common example of weights $W$ is the IPW as discussed in Zhang & Long (2021). However, the inversion of probabilities could cause instability and poor finite-sample performance (Kang & Schafer, 2007). Instead, covariate balancing weights have been proposed as a more stable alternative in causal inference problems (e.g.,

Qin & Zhang, 2007; Imai & Ratkovic, 2014). To motivate the proposed covariate balancing weights in our problem, we decompose the weighted estimator in the following way:

$$\tilde{\mathcal{E}}_a(g, W) = \frac{1}{n}\sum_{i=1}^{n} S_i W_i \mathbb{I}(A_i = a)\left\{\ell(g; Z_i) - r(g, U_i, a)\right\} \tag{4}$$

$$+ \frac{1}{n}\sum_{i=1}^{n} S_i W_i \mathbb{I}(A_i = a) r(g, U_i, a) - \frac{1}{n_a}\sum_{j=1}^{n} \mathbb{I}(A_j = a) r(g, U_j, a) \tag{5}$$

$$+ \frac{1}{n_a}\sum_{j=1}^{n} \mathbb{I}(A_j = a) r(g, U_j, a) - \mathbf{E}_{\mathcal{T}}\{r(g, U, a)|A = a\} \tag{6}$$

$$+ \mathbf{E}_{\mathcal{T}}\{r(g, U, a)|A = a\},$$

where $n_a = \sum_{i=1}^{n} \mathbb{I}(A_i = a)$. Note that $\mathbf{E}_{\mathcal{T}}\{r(g, U, a)|A = a\} = \mathbf{E}_{\mathcal{T}}\{\ell(g; Z)|A = a\} = \mathcal{E}_a(g)$. Therefore, we want the magnitude of the three terms (4)–(6) small. For large samples, the terms (4) and (6) are expected to be small due to concentration arguments, similarly as in prior works on covariate balancing weights (e.g., Wong & Chan, 2018; Wang & Zubizarreta, 2019). The key is to find weights that control the term (5), and therefore our goal is to find weights $\{W_i\}$ to control

$$\left|\frac{1}{n}\sum_{i=1}^{n} S_i W_i \mathbb{I}(A_i = a) r(g, U_i, a) - \frac{1}{n_a}\sum_{j=1}^{n} \mathbb{I}(A_j = a) r(g, U_j, a)\}\right|$$

$$= \left|\sum_{i=1}^{n} \mathbb{I}(A_i = a)\left(\frac{S_i W_i}{n} - \frac{1}{n_a}\right) r(g, U_i, a)\right|, \tag{7}$$

which is the absolute difference between the weighted conditional loss of function $g$ over $p_{\mathcal{T}}$ using the observed data and the corresponding average conditional loss. However, an issue shows up immediately. The conditional expectation $r$ is not known in general, rendering (7) inaccessible. Moreover, the optimization associated with the fair regression using (3) will be highly complicated if the weights depend on $g$. To solve these issues, we instead control the supreme (uniform) error over a large class of functions of $U$ without specific choices of $g$ or $r$. The underlying idea is that if we can control the supreme (uniform) error

$$\sup_{h \in \mathcal{F}}\left|\sum_{i=1}^{n} \mathbb{I}(A_i = a)\left(\frac{S_i W_i}{n} - \frac{1}{n_a}\right) h(U_i, a)\right| \tag{8}$$

such that the function class $\mathcal{F}$ includes or approximates $r(g, U, a)$ well, we can control (7). Following a similar uniform balancing idea in Wong & Chan (2018) that simultaneously controls a uniform error and a variability measure of weights, our weights result from a minimax optimization problem, and the corresponding details are present in Appendix A.1. Note that the weights in Wong & Chan (2018) are developed for average treatment estimation, which is a conditional mean (a scalar), and so is different from our target (the risk function). Our weights result from a different minimax optimization problem.

## 4 WEIGHTED FAIR REGRESSION

With the weights introduced in Section 3, we now propose an estimator for the population fair regression function $g_\star$. Let $\hat{W}$ denote the estimated balancing weights by the optimization step in Appendix A.1. We can estimate the risk over $p_{\mathcal{T}}$:

$$\mathbf{E}_{\mathcal{T}}\{\ell(g; Z)\} = \sum_{a=0,1} \mathbf{E}_{\mathcal{T}}\{\ell(g; Z)|A = a\} P(A = a)$$

by

$$\tilde{R}_{\mathcal{T}}(g, \hat{W}) := \sum_{a=0,1} \tilde{\mathcal{E}}_a(g, \hat{W})\frac{n_a}{n} = \frac{1}{n}\sum_{i=1}^{n} S_i \sum_{a=0,1} \mathbb{I}(A_i = a)\frac{\hat{W}_i n_a}{n}\ell(g; Z_i). \tag{9}$$

The proposed estimator of $g_\star$ is the solution to the following minimization:

$$\min_{g \in \mathcal{G}} \left\{ \tilde{R}_\mathcal{T}(g, \hat{W}) + \eta J(g) : |\widetilde{\Delta}(g, \hat{W})| \leq \delta \right\}, \tag{10}$$

where $\widetilde{\Delta}(g, \hat{W})$ and $\tilde{R}_\mathcal{T}(g, \hat{W})$ are as defined in (3) and (9) respectively. Let $J(g)$ denote the complexity of the function and $\eta \geq 0$ denote the tuning parameter.

Note that, given the weights, both the objective function and the constraint can be evaluated based on only the fully observed data, i.e., $\{X_i, Y_i : S_i = 1\}$. As such, we rewrite the optimization using this set of samples. Without loss of generality, we assume that $S_i = 1$ for $i \leq m$ and $S_i = 0$ for $i > m$. Let $\boldsymbol{X} = (X_1, \ldots, X_m)^T$ denote the $m \times p$ design matrix, and $\boldsymbol{Y} = (Y_1, \ldots, Y_m)^T$ denote the vector of response variable. Let $\boldsymbol{W}$ denote a $m \times m$ diagonal matrix where $\boldsymbol{W}_{ii} = \sum_{a=0}^1 \mathbb{I}(A_i = a) \frac{n_a \hat{W}_i}{n^2}$. Let $\boldsymbol{D}$ denote a $m \times m$ diagonal matrix with $\boldsymbol{D}_{ii} = \mathbb{I}(A_i = 1)\frac{\hat{W}_i}{n} - \mathbb{I}(A_i = 0)\frac{\hat{W}_i}{n}$.

**Fair ridge regression**   We first consider a linear regression model class $\{g \in \mathcal{G} : g(X) = X^T \boldsymbol{\beta}\}$ and a ridge penalty $J(g) = ||\boldsymbol{\beta}||_2^2$. Then, the optimization can be written as

$$\min_{\boldsymbol{\beta}} \left\{ (\boldsymbol{Y} - \boldsymbol{X}\boldsymbol{\beta})^T \boldsymbol{W} (\boldsymbol{Y} - \boldsymbol{X}\boldsymbol{\beta}) + \eta \boldsymbol{\beta}^T \boldsymbol{\beta} : |(\boldsymbol{Y} - \boldsymbol{X}\boldsymbol{\beta})^T \boldsymbol{D} (\boldsymbol{Y} - \boldsymbol{X}\boldsymbol{\beta})| \leq \delta \right\}. \tag{11}$$

**Fair kernel ridge regression**   Next, we consider a nonparametric alternative: a fair version of kernel ridge regression. Consider some reproducing kernel Hilbert space (RKHS) $\mathcal{H}$ with a reproducing kernel function $\boldsymbol{R}(\cdot, \cdot)$. With $J(g) = ||g||_\mathcal{H}^2$ denoting the complexity, the optimal solution of (10) lies in the space spanned by $\{\boldsymbol{R}(X_1, \cdot), \ldots, \boldsymbol{R}(X_m, \cdot)\}$ due to the representer theorem. Therefore, we focus on a class of functions $\{g \in \mathcal{G} : g(\cdot) = \sum_{i=1}^m \alpha_i \boldsymbol{R}(X_i, \cdot)\}$.

Let $\mathbf{K}$ denote the Gram matrix that $\mathbf{K}_{i,j} = \boldsymbol{R}(X_i, X_j)$ for $1 \leq i, j \leq m$. We can reformulate the optimization problem (10) in terms of $\boldsymbol{\alpha} = (\alpha_1, \ldots, \alpha_m)^T$:

$$\min_{\boldsymbol{\alpha}} \left\{ (\boldsymbol{Y} - \mathbf{K}\boldsymbol{\alpha})^T \boldsymbol{W} (\boldsymbol{Y} - \mathbf{K}\boldsymbol{\alpha}) + \eta \boldsymbol{\alpha}^T \mathbf{K} \boldsymbol{\alpha} : |(\boldsymbol{Y} - \mathbf{K}\boldsymbol{\alpha})^T \boldsymbol{D} (\boldsymbol{Y} - \mathbf{K}\boldsymbol{\alpha})| \leq \delta \right\}. \tag{12}$$

Let $\mathbf{K} = \boldsymbol{L}\boldsymbol{L}^T$. The above optimization (12) can be rewritten as

$$\min_{\boldsymbol{\beta}} \left\{ (\boldsymbol{Y} - \boldsymbol{L}\boldsymbol{\beta})^T \boldsymbol{W} (\boldsymbol{Y} - \boldsymbol{L}\boldsymbol{\beta}) + \eta \boldsymbol{\beta}^T \boldsymbol{\beta} : |(\boldsymbol{Y} - \boldsymbol{L}\boldsymbol{\beta})^T \boldsymbol{D} (\boldsymbol{Y} - \boldsymbol{L}\boldsymbol{\beta})| \leq \delta \right\}, \tag{13}$$

where $\boldsymbol{\beta}$ is equivalent to $\boldsymbol{L}^T \boldsymbol{\alpha}$. Note that the fair ridge regression (11) can be expressed with the same form in (13) by replacing $\boldsymbol{L}$ with $\boldsymbol{X}$.

In the following subsections, we focus on solving optimization in the form of (13). Note that each of the key term in the fairness constraint $(\boldsymbol{Y} - \boldsymbol{L}\boldsymbol{\beta})^T \boldsymbol{D} (\boldsymbol{Y} - \boldsymbol{L}\boldsymbol{\beta})$ is a difference between two convex functions, and so the resulting optimization is a challenging non-convex optimization problem. We could use the disciplined convex-concave program (DCCP) to solve this optimization (Shen et al., 2016). However, there is no guarantee that DCCP will return the globally optimal solution. Therefore, we next introduce the Lagrangian duality and sufficient conditions to find the global optimal solution by solving the dual problem. We provide comparisons between the DCCP and our optimization method in Appendix B.4. The results demonstrate the proposed algorithm consistently achieves a smaller objective value, and is significantly more efficient, reducing both the average and maximum computation time to less than 1% of DCCP optimization.

### 4.1  LAGRANGIAN DUALITY AND GLOBAL OPTIMAL SOLUTION

In this section, we formalize the constrained optimization problem as a convex problem through the Lagrangian duality.

#### 4.1.1  THE LAGRANGIAN DUAL FORM

The Lagrangian form of (13) can be expressed with one free Lagrange multiplier $\lambda$, or equivalently two nonnegative Lagrange multipliers

$$\lambda_+ := \max\{\lambda, 0\}, \lambda_- := -\min\{\lambda, 0\}, \lambda = (\lambda_+ - \lambda_-), \tag{14}$$

as follows:

$$L(\boldsymbol{\beta}, \lambda_+, \lambda_-) = \boldsymbol{\beta}^T(\boldsymbol{L}^T(\boldsymbol{W} - \lambda\boldsymbol{D})\boldsymbol{L} + \eta\mathbf{I})\boldsymbol{\beta} - 2\boldsymbol{Y}^T(\boldsymbol{W} - \lambda\boldsymbol{D})\boldsymbol{L}\boldsymbol{\beta}$$
$$+ \lambda_+(-\boldsymbol{Y}^T\boldsymbol{D}\boldsymbol{Y} - \delta) - \lambda_-(-\boldsymbol{Y}^T\boldsymbol{D}\boldsymbol{Y} + \delta). \tag{15}$$

Let $p^*$ denote the solution to the primal problem (13). It can be expressed as the solution to a minimax optimization of the Lagrangian form. However, solving a minimax optimization of the Lagrangian form is usually hard, hence many primal-dual approximation algorithms (e.g. Coston et al., 2019) instead solve the dual problem. The dual problem of the optimization in (13), which is the maximin optimization of the Lagrangian form, is written as:

$$d^* := \sup_{\lambda} \inf_{\boldsymbol{\beta}} L(\boldsymbol{\beta}, \lambda_+, \lambda_-)$$
$$= \sup_{\lambda} -(\boldsymbol{L}^T(\boldsymbol{W} - \lambda\boldsymbol{D})\boldsymbol{Y})^T(\boldsymbol{L}^T(\boldsymbol{W} - \lambda\boldsymbol{D})\boldsymbol{L} + \eta\mathbf{I})^-(\boldsymbol{L}^T(\boldsymbol{W} - \lambda\boldsymbol{D})\boldsymbol{Y})$$
$$+ \lambda_+(-\boldsymbol{Y}^T\boldsymbol{D}\boldsymbol{Y} - \delta) - \lambda_-(-\boldsymbol{Y}^T\boldsymbol{D}\boldsymbol{Y} + \delta), \tag{16}$$
$$\text{subject to} \begin{cases} \boldsymbol{L}^T(\boldsymbol{W} - \lambda\boldsymbol{D})\boldsymbol{L} + \eta\mathbf{I} \succeq 0 \\ \boldsymbol{L}^T(\boldsymbol{W} - \lambda\boldsymbol{D})\boldsymbol{Y} \in \text{Range}(\boldsymbol{L}^T(\boldsymbol{W} - \lambda\boldsymbol{D})\boldsymbol{L} + \eta\mathbf{I}) \end{cases}.$$

### 4.1.2 THE STRONG DUALITY

Based on the well-known weak duality, a lower bound for $p^*$ is obtained by solving the dual problem. We next present the sufficient conditions for strong duality to hold between the primal form (13) and dual form (16). When such strong duality holds, we can optimize the dual to obtain a globally optimal solution of the primal. Next, we present an assumption and a lemma, which can be shown using Theorem 7 of Wang & Xia (2014).

**Assumption 2.**

$(i)$ $\boldsymbol{L}^T\boldsymbol{D}\boldsymbol{L} \neq 0$.

$(ii)$ *There exists a* $\bar{\boldsymbol{\beta}} \in \mathbb{R}^d$, *such that* $-\delta < (\boldsymbol{Y} - \boldsymbol{L}\bar{\boldsymbol{\beta}})^T\boldsymbol{D}(\boldsymbol{Y} - \boldsymbol{L}\bar{\boldsymbol{\beta}}) < \delta$.

$(iii)$ *The primal problem is bounded below.*

In a weighted regression problem, Assumption 2(i) is a mild assumption. Given that $\boldsymbol{L}$ and $\boldsymbol{D}$ denote the design matrix and weight matrix, respectively, this assumption is violated only if the group difference between the weighted sum of products of any two columns of $\boldsymbol{L}$ equals zero, which is generally not the case. Assumption 2(ii) posits that there always exists an interior point within the feasible region. In Appendix A.2, it can be shown that this assumption holds for the regression problem under certain general conditions. Assumption 2(iii) always holds for regression problem.

**Lemma 1** (Strong Duality). *Under Assumption 2$(i)$ and 2$(ii)$,*

$$d^* := \sup_{\lambda} \inf_{\boldsymbol{\beta}} L(\boldsymbol{\beta}, \lambda_+, \lambda_-) = \inf_{\boldsymbol{\beta}} \sup_{\lambda} L(\boldsymbol{\beta}, \lambda_+, \lambda_-) =: p^* \tag{17}$$

*Moreover, if Assumption 2$(iii)$ holds in addition to 2$(i)$ and 2$(ii)$, $d^*$ is attained.*

### 4.1.3 THE CONDITIONS TO CHARACTERIZE A GLOBAL OPTIMAL SOLUTION

In this section, we introduce a proposition to characaterize the global optimal solution to the primal (13). By Theorem 2.4 of Pong & Wolkowicz (2014), there exists a nonempty open interval $(\underline{\lambda}, \overline{\lambda})$ containing 0, such that $\boldsymbol{L}^T(\boldsymbol{W} - \lambda\boldsymbol{D})\boldsymbol{L} + \eta\mathbf{I} \succeq 0$ if and only if $\underline{\lambda} \leq \lambda \leq \overline{\lambda}$, and $\boldsymbol{L}^T(\boldsymbol{W} - \lambda\boldsymbol{D})\boldsymbol{L} + \eta\mathbf{I} \succ 0$ if and only if $\underline{\lambda} < \lambda < \overline{\lambda}$. The characterization of $\underline{\lambda}$ and $\overline{\lambda}$ can be found in Appendix A.3. For a fixed $\lambda \in \text{Closure}(\underline{\lambda} : \overline{\lambda})$, we define the first-order stationary point of the Lagrangian form (15) as,

$$\boldsymbol{\beta}(\lambda) := (\boldsymbol{L}^T(\boldsymbol{W} - \lambda\boldsymbol{D})\boldsymbol{L} + \eta\mathbf{I})^-\boldsymbol{L}^T(\boldsymbol{W} - \lambda\boldsymbol{D})\boldsymbol{Y}, \tag{18}$$

and the constraint evaluated at $\boldsymbol{\beta}(\lambda)$ as,

$$\psi(\lambda) := \boldsymbol{\beta}(\lambda)^T\boldsymbol{L}^T\boldsymbol{D}\boldsymbol{L}\boldsymbol{\beta}(\lambda) - 2\boldsymbol{Y}^T\boldsymbol{D}\boldsymbol{L}\boldsymbol{\beta}(\lambda)$$
$$= (\boldsymbol{L}^T(\boldsymbol{W} - \lambda\boldsymbol{D})\boldsymbol{Y})^T(\boldsymbol{L}^T(\boldsymbol{W} - \lambda\boldsymbol{D})\boldsymbol{L} + \eta\mathbf{I})^-\boldsymbol{L}^T\boldsymbol{D}\boldsymbol{L}^T(\boldsymbol{L}^T(\boldsymbol{W} - \lambda\boldsymbol{D})\boldsymbol{L} + \eta\mathbf{I})^- \tag{19}$$
$$\times \boldsymbol{L}^T(\boldsymbol{W} - \lambda\boldsymbol{D})\boldsymbol{Y} - 2\boldsymbol{Y}^T\boldsymbol{D}\boldsymbol{L}(\boldsymbol{L}^T(\boldsymbol{W} - \lambda\boldsymbol{D})\boldsymbol{L} + \eta\mathbf{I})^-\boldsymbol{L}^T(\boldsymbol{W} - \lambda\boldsymbol{D})\boldsymbol{Y}.$$

Let $\lambda^*$ denote the optimal Lagrangian multiplier of the dual problem (16), and $\boldsymbol{\beta}^*$ denote the optimal solution of the primal (13). Then the following proposition characterizes $\lambda^*$ and $\boldsymbol{\beta}^*$ under Assumption 2.

**Proposition 1.** *Under Assumption 2, if $-\boldsymbol{Y}^T\boldsymbol{D}\boldsymbol{Y} - \delta \leq \psi(0) \leq -\boldsymbol{Y}^T\boldsymbol{D}\boldsymbol{Y} + \delta$, then $\lambda^* = 0$. Otherwise, $\lambda^*$ is the unique maximizer of the following concave objective function:*

$$h_C(\lambda) = -(\boldsymbol{L}^T(\boldsymbol{W} - \lambda\boldsymbol{D})\boldsymbol{Y})^T(\boldsymbol{L}^T(\boldsymbol{W} - \lambda\boldsymbol{D})\boldsymbol{L} + \eta\mathbf{I})^-(\boldsymbol{L}^T(\boldsymbol{W} - \lambda\boldsymbol{D})\boldsymbol{Y}) + \lambda C,$$

$$\text{subject to} \begin{cases} \boldsymbol{L}^T(\boldsymbol{W} - \lambda\boldsymbol{D})\boldsymbol{L} + \eta\mathbf{I} \succeq 0 \\ \boldsymbol{L}^T(\boldsymbol{W} - \lambda\boldsymbol{D})\boldsymbol{Y} \in \text{Range}(\boldsymbol{L}^T(\boldsymbol{W} - \lambda\boldsymbol{D})\boldsymbol{L} + \eta\mathbf{I}) \end{cases}, \tag{20}$$

*where $C = -\boldsymbol{Y}^T\boldsymbol{D}\boldsymbol{Y} - \delta$ if $\psi(0) < -\boldsymbol{Y}^T\boldsymbol{D}\boldsymbol{Y} - \delta$, and $C = -\boldsymbol{Y}^T\boldsymbol{D}\boldsymbol{Y} + \delta$ if $\psi(0) > -\boldsymbol{Y}^T\boldsymbol{D}\boldsymbol{Y} + \delta$.*

*If $\lambda^* \in (\underline{\lambda}, \overline{\lambda})$, $\boldsymbol{\beta}^* = \boldsymbol{\beta}(\lambda^*)$ is the unique optimizer of the primal; otherwise, there exists a vector $\nu \in \text{Null}(\boldsymbol{L}^T(\boldsymbol{W} - \lambda^*\boldsymbol{D})\boldsymbol{L} + \eta\mathbf{I})\backslash\{0\}$ such that $\boldsymbol{\beta}^* = \boldsymbol{\beta}(\lambda^*) + \nu$.*

See Appendix A.4 for the proof of Proposition 1 and the characterization of $\nu$. This proposition implies that the two inequality constraints in primal (13) can be reduced to a single equality constraint.

### 4.1.4 ALGORITHM

Following Proposition 1, we propose the following algorithm to solve the primal problem (13) by optimizing the Lagrangian multiplier through Newton's method. Our method is different from the extended Rendl-Wolkowicz (ERW) algorithm of Pong & Wolkowicz (2014), which solves the same primal problem as a parametrized eigenvalue problem. We, instead, focus on solving the dual problem in (16) directly. By solving the dual problem, the optimal coefficients, $\boldsymbol{\beta}(\lambda)$, described in (18), will impose a special structure related to $\boldsymbol{Y}$, which is not accessible with the ERW solution. This special structure enables us to perform model selection efficiently, as discussed in the following section.

---

**Algorithm 1** Optimization procedure to solve for problem 13

---

1: Calculate initial interval for $\lambda^*$, $(\underline{\lambda}, \overline{\lambda})$ as in Proposition A.3.
2: **if** $-\boldsymbol{Y}^T\boldsymbol{D}\boldsymbol{Y} - \delta \leq \psi(0) \leq -\boldsymbol{Y}^T\boldsymbol{D}\boldsymbol{Y} + \delta$            ▷ Interior solution **then**
3:     $\lambda^* = 0, \boldsymbol{\beta}^* = \boldsymbol{\beta}(\lambda^*)$
4: **else if** $\psi(0) < -\boldsymbol{Y}^T\boldsymbol{D}\boldsymbol{Y} - \delta$       ▷ Optimal solution reaches lower constraint **then**
5:     **if** $\overline{\lambda} = \infty$ or $\psi(\overline{\lambda}) < -\boldsymbol{Y}^T\boldsymbol{D}\boldsymbol{Y} - \delta$ **then**
6:        Use Newton's method 2 to solve $\lambda^* = \text{argmax}\, h_C(\lambda)$ in (20) for $C = -\boldsymbol{Y}^T\boldsymbol{D}\boldsymbol{Y} - \delta$.
7:        $\boldsymbol{\beta}^* = \boldsymbol{\beta}(\lambda^*)$.
8:     **else**
9:        $\lambda^* = \overline{\lambda}$. Solve for suitable $\nu$ as in (38); $\boldsymbol{\beta}^* = \boldsymbol{\beta}(\lambda^*) + \nu$.
10:     **end if**
11: **else**                     ▷ Optimal solution reaches upper constraint
12:     **if** $\underline{\lambda} = -\infty$ or $\psi(\underline{\lambda}) > -\boldsymbol{Y}^T\boldsymbol{D}\boldsymbol{Y} + \delta$ **then**
13:        Use Newton's method 2 to solve $\lambda^* = \text{argmax}\, h_C(\lambda)$ in (20) for $C = -\boldsymbol{Y}^T\boldsymbol{D}\boldsymbol{Y} + \delta$.
14:        $\boldsymbol{\beta}^* = \boldsymbol{\beta}(\lambda^*)$.
15:     **else**
16:        $\lambda^* = \underline{\lambda}$. Solve for suitable $\nu$ as in (38); $\boldsymbol{\beta}^* = \boldsymbol{\beta}(\lambda^*) + \nu$.
17:     **end if**
18: **end if**
19: **Return** $(\lambda^*, \boldsymbol{\beta}^*)$

---

### 4.2 PARAMETER TUNING

Let $\hat{g}_\eta(\cdot)$ be the predictive function whose complexity depends on $\eta$. A small value of $\eta$ leads to a flexible $\hat{g}_\eta(\cdot)$, and thereby small $\hat{\mathcal{E}}_a(\hat{g}_\eta, \hat{W})$ and small $\widetilde{\Delta}(\hat{g}_\eta, \hat{W})$. However, these models may not generalize well to the out-of-distribution samples as they overfit. To mitigate the effect of overfitting, an appropriate value of $\eta$ has to be selected to control the flexibility of the fitted model. In below, we introduce an approximate leave-one-out-cross-validation (LOOCV) criterion, which does not require re-fittings of the estimator for each $\eta$, and, therefore, will enable fast tuning of $\eta$ (Section 4.2.2).

### 4.2.1 APPROXIMATE GROUP LOOCV

One common way of estimating the generalization error $\mathbf{E}_{\mathcal{T}}\{(Y - \hat{g}_\eta(X))^2\}$ (where the expectation taken over a new observation $(Y, X)$) is through LOOCV (Stone, 1974). Recall $\hat{W}$ is the estimated balancing weights introduced in Section 3. We would like to estimate the conditional risk $\mathbf{E}_{\mathcal{T}}\{(Y - \hat{g}_\eta(X))^2 | A = a\}$ for each group $a$ via a group LOOCV:

$$\hat{\mathcal{E}}_a(\hat{g}_\eta, \hat{W}) = \frac{1}{n} \sum_{i=1}^{n} S_i \mathbb{I}(A_i = a) \hat{W}_i (Y_i - \hat{Y}_{i(-i)})^2, \tag{21}$$

where $\hat{Y}_{i(-i)}$ denote the evaluation of predictive function estimated by excluding the $i$-th observation.

Let $\boldsymbol{L}_{(-i)}$ and $\boldsymbol{Y}_{(-i)}$ denote the covariate matrix and outcome vector with the $i$-th observation $(L_i, Y_i)$ left out. Let $\boldsymbol{D}_{(-i)}$ and $\boldsymbol{W}_{(-i)}$ denote the corresponding $(m-1) \times (m-1)$ weight matrices. Without loss of generality, we consider the following optimizations:

$$\hat{\boldsymbol{\beta}} = \underset{\boldsymbol{\beta}}{\operatorname{argmin}} (\boldsymbol{L}\boldsymbol{\beta} - \boldsymbol{Y})^T \boldsymbol{W} (\boldsymbol{L}\boldsymbol{\beta} - \boldsymbol{Y}) + \eta \boldsymbol{\beta}^T \boldsymbol{\beta},$$
$$\text{subject to } (\boldsymbol{L}\boldsymbol{\beta} - \boldsymbol{Y})^T \boldsymbol{D} (\boldsymbol{L}\boldsymbol{\beta} - \boldsymbol{Y}) = -\delta, \tag{22}$$

$$\hat{\boldsymbol{\beta}}_{(-i)} = \underset{\boldsymbol{\beta}}{\operatorname{argmin}} (\boldsymbol{L}_{(-i)}\boldsymbol{\beta} - \boldsymbol{Y}_{(-i)})^T \boldsymbol{W}_{(-i)} (\boldsymbol{L}_{(-i)}\boldsymbol{\beta} - \boldsymbol{Y}_{(-i)}) + \eta \boldsymbol{\beta}^T \boldsymbol{\beta},$$
$$\text{subject to } (\boldsymbol{L}_{(-i)}\boldsymbol{\beta} - \boldsymbol{Y}_{(-i)})^T \boldsymbol{D}_{(-i)} (\boldsymbol{L}_{(-i)}\boldsymbol{\beta} - \boldsymbol{Y}_{(-i)}) = -\delta. \tag{23}$$

**Proposition 2.** *Let $\underline{\lambda}$ and $\overline{\lambda}$ be the bounds for the Lagrangian multiplier $\lambda$ of* (22) *as defined in Section 4.1.3. Suppose the optimal solutions of the above optimizations,* (22) *and* (23)*, take the following forms*

$$\hat{\boldsymbol{\beta}} = \boldsymbol{\beta}(\lambda) = (\boldsymbol{L}^T(\boldsymbol{W} - \lambda\boldsymbol{D})\boldsymbol{L} + \eta\mathbf{I})^{-1}\boldsymbol{L}^T(\boldsymbol{W} - \lambda\boldsymbol{D})\boldsymbol{Y}, \tag{24}$$

*for some $\lambda = \lambda^* \in (\underline{\lambda}, \overline{\lambda})$ and*

$$\hat{\boldsymbol{\beta}}_{(-i)} = \boldsymbol{\beta}_{(-i)}(\lambda) = (\boldsymbol{L}_{(-i)}^T(\boldsymbol{W}_{(-i)} - \lambda\boldsymbol{D}_{(-i)})\boldsymbol{L}_{(-i)} + \eta\mathbf{I})^{-1}\boldsymbol{L}_{(-i)}^T(\boldsymbol{W}_{(-i)} - \lambda\boldsymbol{D}_{(-i)})\boldsymbol{Y}_{(-i)},$$

*for some $\lambda = \lambda_{(-i)} \in (\underline{\lambda}, \overline{\lambda})$. Then we have*

$$\hat{\boldsymbol{Y}}_{(i)} = \mathbf{H}(\lambda_{(-i)})\boldsymbol{Y}_{(i)}, \tag{25}$$

*where*
$$\mathbf{H}(\lambda) = \boldsymbol{L}(\boldsymbol{L}^T(\boldsymbol{W} - \lambda\boldsymbol{D})\boldsymbol{L} + \eta\mathbf{I})^{-1}\boldsymbol{L}^T(\boldsymbol{W} - \lambda\boldsymbol{D}), \tag{26}$$

*and $\boldsymbol{Y}_{(i)} = (Y_1, \ldots, Y_{i-1}, \hat{Y}_{i(-i)}, Y_{i+1}, \ldots, Y_m)^T$, with $\hat{Y}_{i(-i)} = L_i^T \hat{\boldsymbol{\beta}}_{(-i)}$.*

The proof of the above proposition can be found in Appendix A.5. We can then use $\lambda^*$ to approximate each $\lambda_{(-i)}$, assuming the maximizer of the dual of 23 stays close to $\lambda^*$ when $Y_i$ is substituted by its leave-one-out prediction $\hat{Y}_{i(-i)}$. We denote $\mathbf{H} = \mathbf{H}(\lambda^*)$. With (25), $\hat{Y}_{i(-i)}$ is approximated by $\mathbf{H}_{i,i}\hat{Y}_{i(-i)} + \sum_{j \neq i} \mathbf{H}_{i,j}Y_j$. With the identity $\hat{Y}_i = \mathbf{H}_{i,i}Y_i + \sum_{j \neq i} \mathbf{H}_{i,j}Y_j$, it follows that we can approximate $Y_i - \hat{Y}_{i(-i)}$ with $(Y_i - \hat{Y}_i)/(1 - \mathbf{H}_{i,i})$. Therefore, given a fixed $\eta$, the approximate group LOOCV can be simplified to the following:

$$\hat{\mathcal{E}}_a(\hat{g}_\eta, \hat{W}) \approx \frac{1}{n} \sum_{i=1}^{n} S_i \mathbb{I}(A_i = a) \hat{W}_i \left( \frac{Y_i - \hat{Y}_i}{1 - \mathbf{H}_{i,i}} \right)^2. \tag{27}$$

When $\eta$ becomes unexpectedly large, the assumption of Proposition 2 may not be satisfied if either of $\lambda^*$ or $\lambda_{(-i)}$, the optimizer of the corresponding duals of (22) and (23), hits the bound $\overline{\lambda}$ or $\underline{\lambda}$. Based on Proposition 1, under this scenario, $\boldsymbol{\beta}^*$ is written as $\boldsymbol{\beta}(\lambda) + \nu$ for some vector $\nu$, and the approximate LOOCV may not serve as an accurate estimator of the out-of-sample MSE. However, note that, in both simulations and benchmark experiments, we have not observed this case for a wide range of $\eta$'s. Also, the model may become overly constrained for unexpectedly large $\eta$, often leading to a lack of predictive power. Therefore, model selection is practically unnecessary at this point.

### 4.2.2 TUNING PROCEDURE

For each fixed $\eta$, with the approximate LOOCV in (27), we estimate the out of sample unfairness via

$$\widehat{\Delta}(\hat{g}_\eta, \hat{W}) = \left| \hat{\mathcal{E}}_1(\hat{g}_\eta, \hat{W}) - \hat{\mathcal{E}}_0(\hat{g}_\eta, \hat{W}) \right|,$$

and the MSE via

$$\hat{R}_\mathcal{T}(\hat{g}_\eta, \hat{W}) = \frac{n_1}{n} \hat{\mathcal{E}}_1(\hat{g}_\eta, \hat{W}) + \frac{n_0}{n} \hat{\mathcal{E}}_0(\hat{g}_\eta, \hat{W}).$$

For a given $\delta$, we select $\eta$ that corresponds to the smallest $\hat{R}_\mathcal{T}(\hat{g}_\eta, \hat{W})$ among all values of $\eta$ such $\widehat{\Delta}(\hat{g}_\eta, \hat{W}) \leq k\delta$, where $k \geq 1$ is a hyperparameter. If there is no value of $\eta$ that satisfies $\widehat{\Delta}(\hat{g}_\eta, \hat{W}) \leq k\delta$, we select $\eta$ corresponding to the smallest $\hat{R}_\mathcal{T}(\hat{g}_\eta, \hat{W})$. We set $k = 1.2$ for all our numerical experiments.

## 5 EXPERIMENTAL EVALUATION

We compare our estimator combined with various weighting techniques, as well as the method proposed in Du et al. (2022) to adjust for selection bias. As Du et al. (2022) specifically addresses the scenario where only the response variable $(Y)$ in training data is affected by selection, we conduct our experiments under the same condition. To better illustrate the effectiveness of our method, we conduct numerical simulations to evaluate our method under various $\delta$ and selection mechanisms for both parametric and nonparametric regressions. As accuracy disparity in the population distribution can arise without selection bias, we would also like to assess our method's capacity of regulating accuracy disparty under this scenario. Additionally, we compare our method with the CENet and WassersteinNet methods proposed in Chi et al. (2021), both of which regulate the MSE disparity for regression model under a setting without bias selection. Due to the space limit, we refer to Appendix B.3 and B.2 for the full experiment details and results of both numerical simulation and comparisons with Chi et al. (2021).

**Dataset**   We carry out the experiments using the Law School Dataset Wightman (1998) and Community&Crime Redmond (2009) Dataset. We adopt the same data generation and preprocessing steps as in Du et al. (2022).[1] In both datasets, a binary variable is selected as a protected attribute $(A)$ and a continuous variable is selected as the outcome $(Y)$. See Appendix 5 for more details about the datasets.

**Method**   We examine the performance of balancing weight adjustment. We further compare it with two inverse probability weights (IPW), proposed in Zhang & Long (2021), where the propensity score (as a function of $U$) is estimated by either logistic regression or kernel support vector machine. The weighted constrained regression is then optimized through the proposed Algorithm 1. Du et al. (2022) employed the Heckman model for bias correction, assuming that the propensity score follows the probit model of a linear function of $U$. They derived the Inverse Mills Ratio (IMR) and treated it as an extra covariate for predicting $Y$. To prevent multicollinearity in predicting $Y$, they further assumed that the predictors set $\tilde{X}$ used to predict $Y$ is a strict subset of $U$, i.e., some covariates not subject to selection are excluded from predicting $Y$ on purpose. Both of these two above assumptions are not required by our method, making our setting more general than the one in Du et al. (2022). Du et al. (2022) handled the constrained optimization by optimizing its Lagrangian dual form while controlling the trade-off by manually tuning the dual variable. In the experiments, we use the default values of the dual variable provided in their code for each dataset.

In the Community&Crime Dataset, the selection mechanism follows:

$$S \mid \text{``NumUnderPov''} \sim \text{Bernoulli}(\Phi(-1.5\text{``NumUnderPov''} + 0.4)),$$

where $\Phi(\cdot)$ represents the cumulative distribution function of the standard normal distribution, and "NumUnderPov" is the ratio of people under the poverty level in a community. We conducted two experiments on this dataset, where the "NumUnderPov" is either included in the set of $U$ or not, which is the same setting used in Du et al. (2022).

---

[1] The source code of implementation, as well as data preprocessing, can be found at https://drive.google.com/file/d/1cOdyTXUeaB1f4lxuyY-bCshUykHOqACA/view.

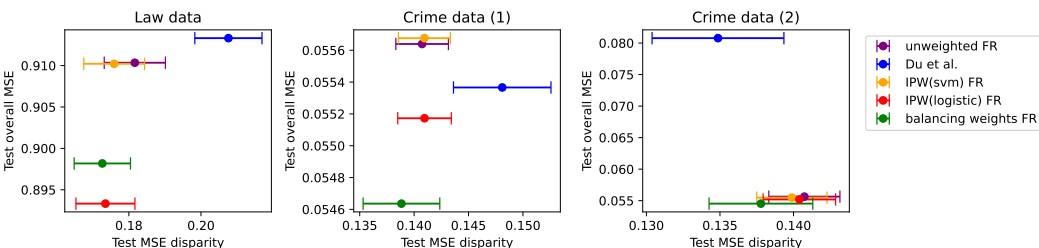

Figure 1: Overall performance: average test MSE and MSE disparity with standard error of different methods on two datasets. "NumUnderPov" attribute is not included in $U$ in the experiment on the Crime data (1).

In the Law School Dataset, the selection mechanism follows:

$$S \mid \text{"age"} \sim \text{Bernoulli}(\text{sigmoid}(-0.2\text{"age"} - 10.5)),$$

where "age" denotes the age of a candidate and is included in $U$.

The specification of all attributes in $\tilde{X}$ and $U$ for both data are discussed in Appendix B.1.

**Results and analysis**   In all experiments, we set constraint $\delta = 0.001$. For each repetition, we randomly split the dataset into 70% training and 30% testing sets. We fit the weighted linear regression model using the restricted covariates set $\tilde{X}$ on the training set (subject to selection) and evaluate its performance on the testing set (not subject to selection). We summarize the average performance of 200 repetitions as well as the standard error for each method in Figure 1. Our observations are as follows: (1) The proposed method achieves a smaller MSE and an MSE disparity at least as small as the method proposed in Du et al. (2022) on all experiments. (2) Among all weights that we have compared, balancing weights achieves the smallest MSE disparity on all experiments. The proposed method also achieves the best trade-off (the average value of the metric is located at the bottom left) on both experiments on Crime data. (3) When "NumUnderPov" is excluded from $U$, Assumption 1 might be violated. However, the experiment results show that balancing weights adjustment is still effective in regulating the MSE disparity on population distribution.

# 6   CONCLUSION

In this paper, we propose a novel weighted fair regression algorithm to regulate the MSE disparity on population distribution given the training data is under selection bias. Our algorithm solves a non-convex constrained optimization problem through its Lagrangian dual, obtaining the globally optimal solution. The efficiency of our proposed algorithm is validated under comprehensive experiments.

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

# A  PROOF AND DERIVATIONS

## A.1  ESTIMATION OF THE BALANCING WEIGHTS $W$ IN SECTION 3

In this section, we aim to control the error term in (7) and denote the uniform weights for controlling the error term for group $a$ as $W_a$.

For each $a = 0, 1$, we would like to obtain the weights $W_a$ such that

$$\sum_{i=1}^{n} \mathbb{I}(A_i = a) \left( \frac{S_i W_{a,i}}{n} - \frac{1}{n_a} \right) r(g, U_i, a) \approx 0, \tag{28}$$

for a large class of $r$ and $g$. As mentioned in Section 3, both $g$ and $r$ are unknown, therefore we use the generic notation $h$ to denote their combined effect and $h(U_i, a)$ as a substitute to $r(U_i, g, a)$, and we aim to control the supreme (uniform) error. Denote

$$H_n(W_a, h, a) = \left\{ \sum_{i=1}^{n} \mathbb{I}(A_i = a) \left( \frac{S_i W_{a,i}}{n} - \frac{1}{n_a} \right) h(U_i, a) \right\}^2,$$

we need to control $H_n(W_a, h, a)$ to validate the balancing weights $W_a$. We have

$$H_n(W_a, h, a) \leq \frac{1}{n} \sum_{i=1}^{n} \mathbb{I}(A_i = a) h^2(U_i, a) \frac{1}{n} \sum_{i=1}^{n} \mathbb{I}(A_i = a) \left( S_i W_{a,i} - \frac{n}{n_a} \right)^2.$$

Therefore, following Wong & Chan (2018), we consider the following constrained minimizations for $a = 0, 1$ for some reproducing-kernel Hilbert space $\mathcal{H}$ with norm $|| \cdot ||_{\mathcal{H}}$ of functions on $U_i$:

$$\min_{W_{a,i} \geq \frac{n}{n_a}} \left[ \sup_{h \in \widetilde{\mathcal{H}}_n} \{ H_n(W_a, h, a) - \lambda_1 ||h||_{\mathcal{H}} \} + \lambda_2 V_n(W_a) \right],$$

where

$$||h||_n^2 = n^{-1} \sum_{i=1}^n \mathbb{I}(A_i = a) h^2(U_i, a),$$

$$\tilde{\mathcal{H}}_n = \{h \in \mathcal{H} : ||h||_n = 1\},$$

$$V_n(W_a) = \frac{1}{n} \sum_{i=1}^n \mathbb{I}(A_i = a) S_i W_{a,i}^2,$$

Let $K$ be the reproducing kernel of $\mathcal{H}$, and $h(U, a)$ lies in a finite-dimensional subspace span $\{K(U_j, \cdot) : j = 1, ..., n; A_j = a\}$. We can then write $h(U, a) = \sum_{j=1, A_j=a}^n \alpha_{a,j} K(U_j, \cdot)$. Let $M$ denote the $n_a \times n_a$ Gram matrix, i.e., $M_{ij} = K(U_{a_i}, U_{a_j}), A_{a_i} = A_{a_j} = a$. We have

$$H_n(W_a, \sum_{j=1, A_j=a}^n \alpha_{a,j} K(U_j, \cdot), a) = \frac{1}{n^2} \alpha_a^T M Q(W_a, a) M \alpha_a,$$

where

$$Q(W_a, a) = q(W_a, a) q(W_a, a)^T \quad \text{with} \quad q(W_a, a) = \left( (S_{a_1} W_{a_1} - \frac{n}{n_a}), ..., (S_{a_{n_a}} W_{a_{n_a}} - \frac{n}{n_a}) \right)^T.$$

Let the eigenvalue decomposition of $M$ be

$$M = (P_1 \quad P_2) \begin{pmatrix} Q_1 & 0 \\ 0 & Q_2 \end{pmatrix} \begin{pmatrix} P_1^T \\ P_2^T \end{pmatrix}.$$

Since the counterpart of $\frac{n_a}{n} W_{a,i}$ is $P(S_i = 1 | U_i, a)$, which is at least 1, we restrict $W_{a,i} \geq \frac{n}{n_a}$. The weights $W_a$ for each $a = \{0, 1\}$ is the solution of the following problem:

$$\min_{W_a \geq \frac{n}{n_a}} \left[ \sigma_{\max} \left\{ \frac{1}{n} P_1 Q(W_a, a) P_1 - n \lambda_1 Q_1^{-1} \right\} + \lambda_2 V_n(W_a) \right],$$

where $\sigma_{\max}(\cdot)$ represents the maximum eigenvalue of a matrix, and $\lambda_1$, $\lambda_2$ are tuning parameters. Finally, the estimated balancing weights $\hat{W}_i$ in Section 3 is given by $\hat{W}_i = W_{a,i}$ for $A_i = a$.

## A.2 A NOTE ON FEASIBILITY

Assumption 2 (ii) posits that there always exists an interior point within the feasible region. In this section, we would like to show that this assumption holds for the regression problem under certain general conditions.

We consider the most extreme case where $\delta = 0$ and $g(\boldsymbol{X}) = c$ is a constant predictive function. We aim to demonstrate that there exists at least one feasible solution under this scenario. It follows that this solution is an interior point of the feasible region for any function $g$ with an intercept term and $\delta > 0$, and therefore Assumption 2 (ii) holds.

Under this most extreme case, a feasible solution of the optimization problem has to satisfy the following constraint:

$$(\boldsymbol{Y} - c)^T \boldsymbol{D} (\boldsymbol{Y} - c) = 0. \tag{29}$$

The constraint can be expanded as

$$(\boldsymbol{Y} - c)^T \boldsymbol{D} (\boldsymbol{Y} - c)$$

$$= \sum_{i=1}^m \mathbb{I}(A_i = 1) \frac{\hat{W}_i}{n} (Y_i - c)^2 - \sum_{i=1}^m \mathbb{I}(A_i = 0) \frac{\hat{W}_i}{n} (Y_i - c)^2$$

$$= c^2 \left( \sum_{i=1}^m \mathbb{I}(A_i = 1) \frac{\hat{W}_i}{n} - \sum_{i=1}^m \mathbb{I}(A_i = 0) \frac{\hat{W}_i}{n} \right) + 2c \left( - \sum_{i=1}^m \mathbb{I}(A_i = 1) \frac{\hat{W}_i}{n} Y_i + \sum_{i=1}^m \mathbb{I}(A_i = 0) \frac{\hat{W}_i}{n} Y_i \right)$$

$$+ \sum_{i=1}^m \mathbb{I}(A_i = 1) \frac{\hat{W}_i}{n} Y_i^2 - \sum_{i=1}^m \mathbb{I}(A_i = 0) \frac{\hat{W}_i}{n} Y_i^2.$$

$$\tag{30}$$

The balancing weight estimation ensured $\sum_{i=1}^m \mathbb{I}(A_i = 1)\frac{\hat{W}_i}{n} = \sum_{i=1}^m \mathbb{I}(A_i = 0)\frac{\hat{W}_i}{n} = 1$ with minor regularity on the optimization step. It then follows that a value of $c$ satisfying the following equation will satisfy the constraint in (29),

$$2c(-\sum_{i=1}^m \mathbb{I}(A_i = 1)\frac{\hat{W}_i}{n}Y_i + \sum_{i=1}^m \mathbb{I}(A_i = 0)\frac{\hat{W}_i}{n}Y_i) + \sum_{i=1}^m \mathbb{I}(A_i = 1)\frac{\hat{W}_i}{n}Y_i^2 - \sum_{i=1}^m \mathbb{I}(A_i = 0)\frac{\hat{W}_i}{n}Y_i^2 = 0$$
(31)

Such $c$ exists whenever $\sum_{i=1}^m \mathbb{I}(A_i = 1)\frac{\hat{W}_i}{n}Y_i \neq \sum_{i=1}^m \mathbb{I}(A_i = 0)\frac{\hat{W}_i}{n}Y_i$, which can be assumed without loss of generality. Hence, we can conclude that Assumption 2 (ii) holds with minor regularity on the optimization step of weight estimation.

A.3   CHARACTERIZATIONS OF BOUNDS FOR LAGRANGIAN MULTIPLIER

Without loss of generality, we assume $\boldsymbol{L}^T\boldsymbol{W}\boldsymbol{L} + \eta\mathbf{I} \succ 0$, then the matrix pencil $\boldsymbol{L}^T\boldsymbol{W}\boldsymbol{L} + \eta\mathbf{I} - \lambda\boldsymbol{L}^T\boldsymbol{D}\boldsymbol{L}$ is positively regular, i.e, there exist some $\hat{\lambda} \in \mathbb{R}$ such that

$$\boldsymbol{L}^T\boldsymbol{W}\boldsymbol{L} + \eta\mathbf{I} - \hat{\lambda}\boldsymbol{L}^T\boldsymbol{D}\boldsymbol{L} \succ 0.$$

With the above fact, it is proven by (Pong & Wolkowicz, 2014, Theorem 2.4) that there exists a nonempty open interval $(\underline{\lambda}, \overline{\lambda})$ such that $\boldsymbol{L}^T(\boldsymbol{W} - \lambda\boldsymbol{D})\boldsymbol{L} + \eta\mathbf{I} \succeq 0$ if and only if $\underline{\lambda} \leq \lambda \leq \overline{\lambda}$, and $\boldsymbol{L}^T(\boldsymbol{W} - \lambda\boldsymbol{D})\boldsymbol{L} + \eta\mathbf{I} \succ 0$ if and only if $\underline{\lambda} < \lambda < \overline{\lambda}$.

We denote $\lambda_{\max}(A, B)$ and $\lambda_{\min}(A, B)$ to be the maximum and minimum generalized eigenvalue of the matrix pair $(A, B)$. Then from Pong & Wolkowicz (2014) and (Nguyen & Ngan Nguyen, 2024, Theorem 1) we have the following proposition that can calculate $\underline{\lambda}$ and $\overline{\lambda}$ efficiently:

**Proposition 3.** *1. If $\boldsymbol{L}^T\boldsymbol{D}\boldsymbol{L} \succ 0$, $\underline{\lambda} = \lambda_{\min}(\boldsymbol{L}^T\boldsymbol{W}\boldsymbol{L} + \eta\mathbf{I}, \boldsymbol{L}^T\boldsymbol{D}\boldsymbol{L}), \overline{\lambda} = \infty$.*

*2. If $\boldsymbol{L}^T\boldsymbol{D}\boldsymbol{L} \prec 0$, $\underline{\lambda} = -\infty, \overline{\lambda} = \lambda_{\max}(\boldsymbol{L}^T\boldsymbol{W}\boldsymbol{L} + \eta\mathbf{I}, \boldsymbol{L}^T\boldsymbol{D}\boldsymbol{L})$.*

*3.   If $\boldsymbol{L}^T\boldsymbol{D}\boldsymbol{L}$ is indefinite, given $\boldsymbol{L}^T\boldsymbol{W}\boldsymbol{L} + \eta\mathbf{I} \succ 0$, $\underline{\lambda} = \frac{1}{\lambda_{\min}(\boldsymbol{L}^T\boldsymbol{D}\boldsymbol{L}, \boldsymbol{L}^T\boldsymbol{W}\boldsymbol{L} + \eta\mathbf{I})}, \overline{\lambda} = -\frac{1}{\lambda_{\min}(-\boldsymbol{L}^T\boldsymbol{D}\boldsymbol{L}, \boldsymbol{L}^T\boldsymbol{W}\boldsymbol{L} + \eta\mathbf{I})}$.*

A.4   PROOF OF PROPOSITION 1

*Proof.* Under Assumption 2, we have the following theorem to characterize the necessary and sufficient conditions for $\boldsymbol{\beta}^*$ being an optimal solution for problem (13).

**Theorem 4** (Theorem 2.3 of Pong & Wolkowicz (2014))**.** *If Assumption 2 holds, $\boldsymbol{\beta}^*$ is a solution to the primal problem* (13) *if and only if for some Lagrangian multiplier $\lambda^* \in \mathbb{R}$, we have*

$$\begin{aligned}
(\boldsymbol{L}^T(\boldsymbol{W} - \lambda^*\boldsymbol{D})\boldsymbol{L} + \eta\mathbf{I})\boldsymbol{\beta}^* &= \boldsymbol{L}^T(\boldsymbol{W} - \lambda^*\boldsymbol{D})\boldsymbol{Y}, \Big\} \textit{ dual fesibility}\\
\boldsymbol{L}^T(\boldsymbol{W} - \lambda^*\boldsymbol{D})\boldsymbol{L} + \eta\mathbf{I} &\succeq 0,\\
-\delta \leq (\boldsymbol{Y} - \boldsymbol{L}\boldsymbol{\beta}^*)^T\boldsymbol{D}(\boldsymbol{Y} - \boldsymbol{L}\boldsymbol{\beta}^*) &\leq \delta, \textit{ primal feasibility}\\
\lambda_+^*(-\delta - (\boldsymbol{Y} - \boldsymbol{L}\boldsymbol{\beta}^*)^T\boldsymbol{D}(\boldsymbol{Y} - \boldsymbol{L}\boldsymbol{\beta}^*)) &= 0, \Big\} \textit{complementary slackness}\\
\lambda_-^*((\boldsymbol{Y} - \boldsymbol{L}\boldsymbol{\beta}^*)^T\boldsymbol{D}(\boldsymbol{Y} - \boldsymbol{L}\boldsymbol{\beta}^*) - \delta) &= 0,
\end{aligned}$$
(32)

Recall the first-order stationary point of the Lagrangian form (15) for $\lambda \in \text{Closure}(\underline{\lambda} : \overline{\lambda})$ is defined as,

$$\boldsymbol{\beta}(\lambda) := (\boldsymbol{L}^T(\boldsymbol{W} - \lambda\boldsymbol{D})\boldsymbol{L} + \eta\mathbf{I})^-\boldsymbol{L}^T(\boldsymbol{W} - \lambda\boldsymbol{D})\boldsymbol{Y},$$
(33)

and the constraint evaluated at $\boldsymbol{\beta}(\lambda)$ is defined as,

$$\begin{aligned}
\psi(\lambda) :=&\boldsymbol{\beta}(\lambda)^T\boldsymbol{L}^T\boldsymbol{D}\boldsymbol{L}\boldsymbol{\beta}(\lambda) - 2\boldsymbol{Y}^T\boldsymbol{D}\boldsymbol{L}\boldsymbol{\beta}(\lambda)\\
=&(\boldsymbol{L}^T(\boldsymbol{W} - \lambda\boldsymbol{D})\boldsymbol{Y})^T(\boldsymbol{L}^T(\boldsymbol{W} - \lambda\boldsymbol{D})\boldsymbol{L} + \eta\mathbf{I})^-\boldsymbol{L}^T\boldsymbol{D}\boldsymbol{L}^T(\boldsymbol{L}^T(\boldsymbol{W} - \lambda\boldsymbol{D})\boldsymbol{L} + \eta\mathbf{I})^-\\
&\boldsymbol{L}^T(\boldsymbol{W} - \lambda\boldsymbol{D})\boldsymbol{Y} - 2\boldsymbol{Y}^T\boldsymbol{D}\boldsymbol{L}(\boldsymbol{L}^T(\boldsymbol{W} - \lambda\boldsymbol{D})\boldsymbol{L} + \eta\mathbf{I})^-\boldsymbol{L}^T(\boldsymbol{W} - \lambda\boldsymbol{D})\boldsymbol{Y}.
\end{aligned}$$
(34)

In the following, we consider three cases dependent on where $\psi(0)$ is located relative to the interval $(-\boldsymbol{Y}^T\boldsymbol{D}\boldsymbol{Y} - \delta, -\boldsymbol{Y}^T\boldsymbol{D}\boldsymbol{Y} + \delta)$. First of all, if $-\boldsymbol{Y}^T\boldsymbol{D}\boldsymbol{Y} - \delta \leq \psi(0) \leq -\boldsymbol{Y}^T\boldsymbol{D}\boldsymbol{Y} + \delta$, then $\lambda^* = 0$

and $\boldsymbol{\beta}^* = \boldsymbol{\beta}(\lambda^*)$ satisfies all the conditions in (32), and therefore $\boldsymbol{\beta}(\lambda^*)$ is the optimal solution of the primal problem according to Theorem 4. Otherwise, we first consider the case $\psi(0) < -\boldsymbol{Y}^T \boldsymbol{D} \boldsymbol{Y} - \delta$. Since $\psi(\lambda)$ is monotonically increasing on $(\underline{\lambda}, \overline{\lambda})$, if $\psi(0) < -\boldsymbol{Y}^T \boldsymbol{D} \boldsymbol{Y} - \delta$, $\lambda^*$ has to be positive in order to satisfy the primal feasibility condition in (32).

Since $\lambda^* > 0$, the objective function in dual problem (16) can be reduced to the following,

$$h_C(\lambda) = -(\boldsymbol{L}^T(\boldsymbol{W} - \lambda \boldsymbol{D})\boldsymbol{Y})^T(\boldsymbol{L}^T(\boldsymbol{W} - \lambda \boldsymbol{D})\boldsymbol{L} + \eta \mathbf{I})^-(\boldsymbol{L}^T(\boldsymbol{W} - \lambda \boldsymbol{D})\boldsymbol{Y}) + \lambda C,$$

$$\text{subject to} \begin{cases} \boldsymbol{L}^T(\boldsymbol{W} - \lambda \boldsymbol{D})\boldsymbol{L} + \eta \mathbf{I} \succeq 0 \\ \boldsymbol{L}^T(\boldsymbol{W} - \lambda \boldsymbol{D})\boldsymbol{Y} \in \text{Range}(\boldsymbol{L}^T(\boldsymbol{W} - \lambda \boldsymbol{D})\boldsymbol{L} + \eta \mathbf{I}) \end{cases}. \tag{35}$$

where $C = -\boldsymbol{Y}^T \boldsymbol{D} \boldsymbol{Y} - \delta$.

Also, from the complementary slackness condition, the primal feasibility condition is reduced to the following:

$$(\boldsymbol{Y} - \boldsymbol{L}\boldsymbol{\beta}^*)^T \boldsymbol{D}(\boldsymbol{Y} - \boldsymbol{L}\boldsymbol{\beta}^*) = -\delta. \tag{36}$$

The derivative of the above objective function $h_C(\lambda)$ is written as

$$-\psi(\lambda) + (-\boldsymbol{Y}^T \boldsymbol{D} \boldsymbol{Y} - \delta).$$

When $\overline{\lambda} = \infty$ or $\psi(\overline{\lambda}) > -\boldsymbol{Y}^T \boldsymbol{D} \boldsymbol{Y} - \delta$, $\psi(\lambda) = -\boldsymbol{Y}^T \boldsymbol{D} \boldsymbol{Y} - \delta$ is solvable within $(\underline{\lambda}, \overline{\lambda})$; it has one unique solution $\lambda^*$ due to the monotonicity of $\psi(\lambda)$. Then $\lambda^*$ is the unique optimizer of $h_C(\lambda)$, and $\boldsymbol{\beta}(\lambda^*)$, which satisfies all conditions in Theorem 4, is the unique optimal solution of the primal problem (13).

When $\overline{\lambda} < \infty$ and $\psi(\overline{\lambda}) \leq -\boldsymbol{Y}^T \boldsymbol{D} \boldsymbol{Y} - \delta$, $\psi(\lambda) = -\boldsymbol{Y}^T \boldsymbol{D} \boldsymbol{Y} - \delta$ is unsolvable within $(\underline{\lambda}, \overline{\lambda})$. Since $-\psi(0) + (-\boldsymbol{Y}^T \boldsymbol{D} \boldsymbol{Y} - \delta) > 0$, we must have $-\psi(\lambda) + (-\boldsymbol{Y}^T \boldsymbol{D} \boldsymbol{Y} - \delta) \geq 0$ for all $\lambda \in (\underline{\lambda}, \overline{\lambda}]$. It follows that $h_C(\lambda)$ is monotonically increasing on $(\underline{\lambda}, \overline{\lambda}]$ and is maximized at $\lambda = \overline{\lambda}$. Therefore, $\lambda^* = \overline{\lambda}$.

Since $\boldsymbol{L}^T(\boldsymbol{W} - \overline{\lambda}\boldsymbol{D})\boldsymbol{L}$ is singular by the construction of $\underline{\lambda}$ and $\overline{\lambda}$ in Appendix A.3, any $\boldsymbol{\beta} = \boldsymbol{\beta}(\overline{\lambda}) + \nu$ for some vector $\nu \in \text{Null}(\boldsymbol{L}^T(\boldsymbol{W} - \overline{\lambda}\boldsymbol{D})\boldsymbol{L})$ satisfies the dual feasibility conditions in Theorem 4. If such $\boldsymbol{\beta}$ satisfying the primal feasibility condition (36) for some suitable vector $\nu$, it is the optimal solution of primal problem. In below, we introduce how to compute such $\nu$.

Let $\tilde{\nu}$ denote a normalized vector in $\text{Null}(\boldsymbol{L}^T(\boldsymbol{W} - \overline{\lambda}\boldsymbol{D})\boldsymbol{L})\backslash\{0\}$. The optimal solution $\boldsymbol{\beta}^*$ is written as

$$\boldsymbol{\beta}^* = (\boldsymbol{L}^T(\boldsymbol{W} - \overline{\lambda}\boldsymbol{D})\boldsymbol{L} + \eta \mathbf{I})^- \boldsymbol{L}^T(\boldsymbol{W} - \overline{\lambda}\boldsymbol{D})\boldsymbol{Y} + \alpha \tilde{\nu},$$

such that $\boldsymbol{\beta}^*$ satisfies the primal feasibility condition (36),

$$\boldsymbol{\beta}^{*T} \boldsymbol{L}^T \boldsymbol{D} \boldsymbol{L}^T \boldsymbol{\beta}^* - 2\boldsymbol{Y}^T \boldsymbol{D} \boldsymbol{L} \boldsymbol{\beta}^* = -\boldsymbol{Y}^T \boldsymbol{D} \boldsymbol{Y} - \delta. \tag{37}$$

By plugging in the form of $\boldsymbol{\beta}^*$, solving (37) is equivalent to solving for a $\alpha$ for the following quadratic equation,

$$a\alpha^2 - 2b\alpha + c = 0, \tag{38}$$

where

$a = \tilde{\nu}^T(\boldsymbol{L}^T \boldsymbol{D} \boldsymbol{L})\tilde{\nu}$,

$b = \tilde{\nu}^T(\boldsymbol{L}^T \boldsymbol{D} \boldsymbol{L}(\boldsymbol{L}^T(\boldsymbol{W} - \overline{\lambda}\boldsymbol{D})\boldsymbol{L} + \eta \mathbf{I})^- \boldsymbol{L}^T(\boldsymbol{W} - \overline{\lambda}\boldsymbol{D})\boldsymbol{Y} - \boldsymbol{Y}^T \boldsymbol{D} \boldsymbol{L})$,

$c = \boldsymbol{Y}^T(\boldsymbol{W} - \overline{\lambda}\boldsymbol{D})\boldsymbol{L}(\boldsymbol{L}^T(\boldsymbol{W} - \overline{\lambda}\boldsymbol{D})\boldsymbol{L} + \eta \mathbf{I})^-(\boldsymbol{L}^T \boldsymbol{D} \boldsymbol{L})(\boldsymbol{L}^T(\boldsymbol{W} - \lambda \boldsymbol{D})\boldsymbol{L} + \eta \mathbf{I})^- \boldsymbol{L}^T(\boldsymbol{W} - \overline{\lambda}\boldsymbol{D})\boldsymbol{Y}$
$- 2\boldsymbol{Y}^T \boldsymbol{D} \boldsymbol{L}(\boldsymbol{L}^T(\boldsymbol{W} - \overline{\lambda}\boldsymbol{D})\boldsymbol{L} + \eta \mathbf{I})^- \boldsymbol{L}^T(\boldsymbol{W} - \overline{\lambda}\boldsymbol{D})\boldsymbol{Y} + \boldsymbol{Y}^T \boldsymbol{D} \boldsymbol{Y} + \delta,$

$$\tag{39}$$

According to Lemma 2.5 in Pong & Wolkowicz (2014), when $\overline{\lambda} < \infty$, $a > 0$ for all $\tilde{\nu} \in \text{Null}(\boldsymbol{L}^T(\boldsymbol{W} - \overline{\lambda}\boldsymbol{D})\boldsymbol{L})\backslash\{0\}$. At the same time, note that $c = \psi(\overline{\lambda}) + \boldsymbol{Y}^T \boldsymbol{D} \boldsymbol{Y} + \delta < 0$, which then guarantees the existence of real solutions $\alpha$ to (38). We let $\alpha$ to be the real solution with a smaller magnitude, i.e,

$$\alpha = \frac{c}{-b - \text{sign(b)}\sqrt{b^2 - ac}}.$$

The vector $\nu$ in Proposition 1 is then given by $\alpha\tilde{\nu}$.

The case where $\psi(0) > -\boldsymbol{Y}^T \boldsymbol{D} \boldsymbol{Y} + \delta$ can be derived in analogy to the above analysis by reducing the objective function of the dual problem 16 to the same expression in (35) with $C = -\boldsymbol{Y}^T \boldsymbol{D} \boldsymbol{Y} + \delta$.

$\square$

## A.5 PROOFS OF PROPOSITION 2

*Proof.* Let $\boldsymbol{L}_{(-i)}$ and $\boldsymbol{Y}_{(-i)}$ denote the covariate matrix and outcome vector with the $i$-th observation $(L_i, \boldsymbol{Y}_i)$ left out. Let $\boldsymbol{D}_{(-i)}$ and $\boldsymbol{W}_{(-i)}$ denote the corresponding $(m-1) \times (m-1)$ weight matrices.

To prove the position, it suffices to show that $\hat{\boldsymbol{\beta}}_{(-i)}$ is also the optimal solution of the following constraint optimization,

$$\underset{\boldsymbol{\beta}}{\arg\min}(\boldsymbol{L}\boldsymbol{\beta} - \boldsymbol{Y}_{(i)})^T \boldsymbol{W}(\boldsymbol{L}\boldsymbol{\beta} - \boldsymbol{Y}_{(i)}) + \eta \boldsymbol{\beta}^T \boldsymbol{\beta} : (\boldsymbol{L}\boldsymbol{\beta} - \boldsymbol{Y}_{(i)})^T \boldsymbol{D}(\boldsymbol{L}\boldsymbol{\beta} - \boldsymbol{Y}_{(i)}) = -\delta, \quad (40)$$

where $\boldsymbol{Y}_{(i)} = (Y_1, \dots, Y_{i-1}, \hat{Y}_{i(-i)}, Y_{i+1}, \dots, Y_m)^T$, with $\hat{Y}_{i(-i)} = L_i^T \hat{\boldsymbol{\beta}}_{(-i)}$.

Firstly, note the Lagrangian form of above optimization is written as

$$\tilde{L}(\boldsymbol{\beta}, \lambda) = \boldsymbol{\beta}^T \boldsymbol{L}^T (\boldsymbol{W} - \lambda \boldsymbol{D})\boldsymbol{L}\boldsymbol{\beta} - 2\boldsymbol{Y}_{(i)}^T (\boldsymbol{W} - \lambda \boldsymbol{D})\boldsymbol{L}\boldsymbol{\beta} + \eta \boldsymbol{\beta}^T \boldsymbol{\beta} + \boldsymbol{Y}_{(i)}^T \boldsymbol{W} \boldsymbol{Y}_{(i)} + \lambda(-\boldsymbol{Y}_{(i)}^T \boldsymbol{D} \boldsymbol{Y}_{(i)} - \delta). \quad (41)$$

We write the first-order stationary point of (41) as

$$\tilde{\boldsymbol{\beta}}(\lambda) = (\boldsymbol{L}^T (\boldsymbol{W} - \lambda \boldsymbol{D})\boldsymbol{L} + \eta \boldsymbol{I})^{-1} \boldsymbol{L}^T (\boldsymbol{W} - \lambda \boldsymbol{D})\boldsymbol{Y}_{(i)}.$$

By definition, $\hat{\boldsymbol{\beta}}_{(-i)}$ is the minimizer of the Lagrangian form corresponding to the leave-one-out primal problem 23 at $\lambda = \lambda_{(-i)}$, i.e.,

$$\hat{\boldsymbol{\beta}}_{(-i)} = \underset{\boldsymbol{\beta}}{\arg\min} \sum_{j \neq i} (\boldsymbol{W}_{jj} - \lambda_{(-i)} \boldsymbol{D}_{jj})(L_j^T \boldsymbol{\beta} - Y_j)^2 + \eta \boldsymbol{\beta}^T \boldsymbol{\beta} + \lambda_{(-i)}(-\delta).$$

Given $\hat{Y}_{i(-i)} = L_i^T \hat{\boldsymbol{\beta}}_{(-i)}$, it follows that

$$\hat{\boldsymbol{\beta}}_{(-i)} = \underset{\boldsymbol{\beta}}{\arg\min} \sum_{j \neq i} (\boldsymbol{W}_{jj} - \lambda_{(-i)} \boldsymbol{D}_{jj})(L_j^T \boldsymbol{\beta} - Y_j)^2 + (\boldsymbol{W}_{ii} - \lambda_{(-i)} \boldsymbol{D}_{ii})(L_i^T \boldsymbol{\beta} - \hat{Y}_{i(-i)})^2 + \eta \boldsymbol{\beta}^T \boldsymbol{\beta} + \lambda_{(-i)}(-\delta).$$

Therefore, $\hat{\boldsymbol{\beta}}_{(-i)}$ is also the minimizer of $\tilde{L}(\boldsymbol{\beta}, \lambda)$ in (41) at $\lambda = \lambda_{(-i)}$, and we can write $\hat{\boldsymbol{\beta}}_{(-i)} = \tilde{\boldsymbol{\beta}}(\lambda_{(-i)})$. Next we want to show $\sup_\lambda \inf_{\boldsymbol{\beta}} \tilde{L}(\boldsymbol{\beta}, \lambda)$ is actually attained at $\lambda = \lambda_{(-i)}$.

The dual problem of (40) is

$$\tilde{h}(\lambda) = -(\boldsymbol{L}^T (\boldsymbol{W} - \lambda \boldsymbol{D})\boldsymbol{Y}_{(i)})^T (\boldsymbol{L}^T (\boldsymbol{W} - \lambda \boldsymbol{D})\boldsymbol{L} + \eta \boldsymbol{I})^- (\boldsymbol{L}^T (\boldsymbol{W} - \lambda \boldsymbol{D})\boldsymbol{Y}_{(i)}) + \lambda(-\boldsymbol{Y}_{(i)}^T \boldsymbol{D} \boldsymbol{Y}_{(i)} - \delta),$$

$$\text{subject to} \begin{cases} \boldsymbol{L}^T (\boldsymbol{W} - \lambda \boldsymbol{D})\boldsymbol{L} + \eta \boldsymbol{I} \succeq 0 \\ \boldsymbol{L}^T (\boldsymbol{W} - \lambda \boldsymbol{D})\boldsymbol{Y}_{(i)} \in \text{Range}(\boldsymbol{L}^T (\boldsymbol{W} - \lambda \boldsymbol{D})\boldsymbol{L} + \eta \boldsymbol{I}) \end{cases}. \quad (42)$$

The unique optimizer of the concave function $\tilde{h}(\lambda)$ in (42) solves the following equation

$$\nabla \tilde{h}(\lambda) = -\delta - (\boldsymbol{L}\tilde{\boldsymbol{\beta}}(\lambda) - \boldsymbol{Y}_{(i)})^T \boldsymbol{D}(\boldsymbol{L}\tilde{\boldsymbol{\beta}}(\lambda) - \boldsymbol{Y}_{(i)}) = 0. \quad (43)$$

It can be shown $\nabla \tilde{h}(\lambda_{(-i)}) = 0$ as

$$(\boldsymbol{L}\tilde{\boldsymbol{\beta}}(\lambda_{(-i)}) - \boldsymbol{Y}_{(i)})^T \boldsymbol{D}(\boldsymbol{L}\tilde{\boldsymbol{\beta}}(\lambda_{(-i)}) - \boldsymbol{Y}_{(i)}) \quad (44)$$

$$= (\boldsymbol{L}\hat{\boldsymbol{\beta}}_{(-i)} - \boldsymbol{Y}_{(i)})^T \boldsymbol{D}(\boldsymbol{L}\hat{\boldsymbol{\beta}}_{(-i)} - \boldsymbol{Y}_{(i)}) \quad (45)$$

$$= (\boldsymbol{L}_{(-i)}\hat{\boldsymbol{\beta}}_{(-i)} - \boldsymbol{Y}_{(-i)})^T \boldsymbol{D}_{(i)}(\boldsymbol{L}_{(-i)}\hat{\boldsymbol{\beta}}_{(-i)} - \boldsymbol{Y}_{(-i)}) \quad (46)$$

$$= -\delta, \quad (47)$$

where the first quality follows from $\hat{\boldsymbol{\beta}}_{(-i)} = \tilde{\boldsymbol{\beta}}(\lambda_{(-i)})$, and the second equality follows from the fact $\hat{Y}_{i(-i)} = L_i^T \hat{\boldsymbol{\beta}}_{(-i)}$, and the third equality is ensured by the primal feasibility condition of the $\hat{\boldsymbol{\beta}}_{(-i)}$, which is the optimal solution to (23). Therefore, $\tilde{h}(\lambda)$ in (42) takes its optimal values at $\lambda = \lambda_{(-i)}$.

According to Proposition 1, $\tilde{\boldsymbol{\beta}}(\tilde{\lambda})$ is the globally optimal solution of (40) if $\tilde{\lambda}$ is the optimizer of $\tilde{h}(\lambda)$ for some $\tilde{\lambda} \in (\underline{\lambda}, \overline{\lambda})$. Since $\lambda_{(-i)} \in (\underline{\lambda}, \overline{\lambda})$ by assumption, and $\lambda_{(-i)}$ is the optimizer of $\tilde{h}(\lambda)$, $\tilde{\boldsymbol{\beta}}(\lambda_{(-i)})$ is the globally optimal solution.

$\qquad\qquad\qquad\qquad\qquad\qquad\qquad\qquad\qquad\qquad\qquad\qquad\qquad\qquad\qquad\quad$ $\square$

## B  EXPERIMENTS

### B.1  DETAILS AND SETTINGS OF THE EXPERIMENTS UNDER SELECTION BIAS

In this section, we provide the specifications of the protected attribute $A$, the prediction covariates set $\widetilde{X}$ and the fully observed covariate $U$ for the two different datasets used in the experiments.

**Crime data (1)**  Crime dataset Redmond (2009) was collected from the 1990 US Census and contains socio-economic information from 1994 communities. The crime rate of a given community is the target variable ($Y$), and the African American Population Ratio (AAPR) is selected as the protected attribute. Communities with an AAPR exceeding 50% are labeled as protected ($A = 1$), resulting in 219 protected communities and 1775 non-protected ones. As specified in Du et al. (2022), 15 attributes are selected as the prediction covariates set $\widetilde{X}$: "racePctHisp", "agePct12t21", "agePct12t29", "agePct16t24", "agePct65up", "numbUrban", "pctUrban", "medIncome", "pctWWage", "pctWFarm-Self", "pctWInvInc", "pctWSocSec", "pctWPubAsst", "pctWRetire", "medFamInc". In addition to $\tilde{X}$, the following 6 attributes are included in $U$ to avoid the multicollinearity issue of Heckman model: "population", "householdsize", "racepctblack", racePctWhite", racePctAsian". After removing attributes with missing values and standardizing all attributes to have zero mean and unit variance, the IMR covariate and weights are learned using the full set of $U$.

**Crime data (2)**  The overall setting is the same as **Crime data (1)**, except the attribute "NumUnder-Pov" is included in this experiment so that Assumption 1 is satisfied.

**Law data**  Law dataset Wightman (1998) was collected from the Law School Admissions Council's National Longitudinal Bar Passage Study, containing personal records of law students who eventually took the bar exam, including their LSAT scores, age, race, and more. The objective is to predict the GPA of a student based on other attributes. We follow the same preprocessing steps for protected attributes and covariates as outlined in Du et al. (2022). The race attribute (black/non-black) is chosen as the protected attribute, with black being considered the protected ($A = 1$) group. Within a total of 20,649 records, 3,000 records are randomly selected for the final dataset, including 1,000 protected and 2,000 non-protected samples. As specified in Du et al. (2022), 5 attributes are included in the prediction covariates set $\widetilde{X}$: "fulltime", "fam_inc", "age", "gender", "pass". In addition to $\widetilde{X}$, the following 6 attributes are included in $U$ but not $\widetilde{X}$ to avoid the multicollinearity issue of Heckman model: "cluster", "lsat", "ugpa", "zgpa". The IMR covariate and all weights are learned using the set of $U$.

### B.2  BENCHMARK EXPERIMENTS WITHOUT SELECTION BIAS

In this section, we validate the performance of our method proposed in Algorithm 1. Additionally, we compare our methods with the CENet method and WassersteinNet method [2] proposed in Chi et al. (2021). Both the CENet and WassersteinNet methods consider MSE disparity as the fairness metric and assume no covariate shift between the training and testing sets. Therefore, to better compare with these two methods, we focus the cases without selection bias, and let $\hat{W}_i = \frac{n_a}{n}$ if $A_i = a$.

We conduct experiments on Law School Wightman (1998), Communities&Crime Redmond (2009) and Medical Insurance Cost Lantz (2013) datasets. For each dataset, a binary variable is selected as the protected attribute ($A$) and a continuous variable is selected as the response variable ($Y$). All experiments in this section follow the same data generation, data preprocessing and covariate sets specification outlined in Chi et al. (2021). We'll provide a brief overview of the general setup,

---

[2]The source code of implementation of both methods can be found at `https://github.com/JFChi/Understanding-and-Mitigating-Accuracy-Disparity-in-Regression`

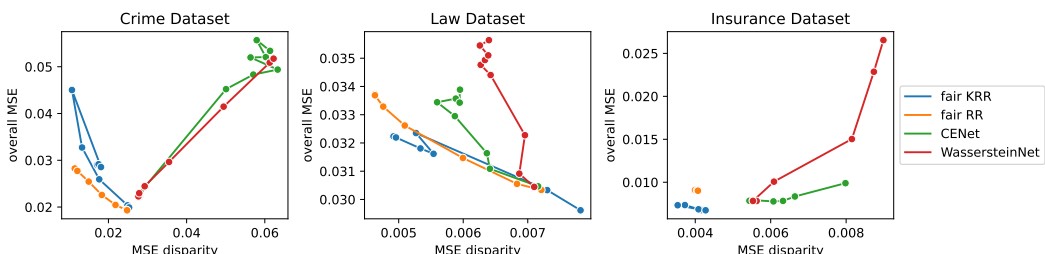

Figure 2: Overall performance: average test MSE and MSE disparity of different methods.

including the protected and target variables. For more detailed information and an introduction to the datasets, readers can refer to the Appendix of Chi et al. (2021).

**Methods** We apply ridge regression (RR) and kernel ridge regression (KRR) with different constraint levels $\delta$ using Lagrangian optimization. We test our method across a wide range of $\delta$ values to understand its performance under various constraints. Both the CENet method and WassersteinNet aim to minimize the MSE disparity of neural network models by fitting them on a fair representation that minimizes the distance between two groups Chi et al. (2021). Algorithmic fairness is controlled by a parameter $\lambda$, which penalizes the distance between conditional representations. In our experiments, we use the default neural network architectures from Chi et al. (2021) and vary $\lambda$ across a wide range of values. Further details on parameter tuning for all methods can be found in the following corresponding paragraphs.

**Crime data** In this experiment, the race attribute is used as the protected attribute with $A = 1$ represents when the population percentage of the white is greater or equal to 80% and 0 otherwise. Each instance has an input dimension of 96. For ridge regression, we set the tuning hyper-parameter range to be 20 numbers spaced evenly on a log scale between $10^{-9}$ and $10^{-3}$. In Kernel ridge regression, we implement the rbf kernel using random Fourier features and choose the bandwidth through median heuristic. We select the penalizing hyper-parameter $\eta$ from 20 evenly spaced numbers on a log scale between $10^{-8}$ and 1. The threshold $\delta$ for our methods is set to be $\{10^{-6}, 10^{-5}, 10^{-4}, 10^{-3}, 0.005, 0.01, 0.015, 0.02, 0.03\}$; the penalizing hyper-parameter $\lambda$ for CENet and WassersteinNet in Chi et al. (2021) is set as $\{0.01, 0.1, 1.0, 5.0, 10.0, 15.0, 20.0, 35.0, 50.0, 100.0\}$.

**Law Data** In this experiment, we use gender as the protected attribute and undergraduate GPA as the target variable. For ridge regression, we set the tuning hyper-parameter range to be 20 numbers spaced evenly on a log scale between $10^{-9}$ and $10^{-3}$. In kernel ridge regression, we use the Sobolev kernel. We select the penalizing hyper-parameter $\eta$ from 20 evenly spaced numbers on a log scale between $10^{-8}$ and 1. The MSE disparity threshold $\delta$ for our methods is set to be $\{10^{-6}, 10^{-5}, 10^{-4}, 10^{-3}, 0.0025, 0.005, 0.075, 0.01\}$; the penalizing hyper-parameter $\lambda$ for CENet and WassersteinNet in Chi et al. (2021) is set as $\{0.01, 0.1, 1.0, 5.0, 10.0, 15.0, 20.0, 50.0, 100.0\}$.

**Medical Insurance Cost Data** The medical insurance cost dataset Lantz (2013) is a simulated dataset created using real-world demographic statistics from the U.S. Census Bureau. It contains 1,338 medical expense examples for patients in the United States. In this experiment, gender is selected as the protected attribute and the charged medical expenses as the target variable. We apply the same sub-sample process as in Chi et al. (2021), by randomly selecting 5% of examples with male gender and 50% of examples with female gender, resulting in a total of 364 examples. For ridge regression, we set the tuning hyper-parameter range to be 20 numbers spaced evenly on a log scale between $10^{-9}$ and $10^{-3}$. We use the Sobolev kernel in kernel ridge regression. We select the penalizing hyper-parameter $\eta$ from 20 evenly spaced numbers on a log scale between $10^{-8}$ and 1. The MSE disparity threshold $\delta$ for our methods is set to be $\{10^{-6}, 10^{-5}, 10^{-4}, 10^{-3}, 0.0025, 0.005, 0.01\}$; the penalizing hyper-parameter $\lambda$ for CENet and WassersteinNet in Chi et al. (2021) is set as $\{0.01, 0.1, 0.5, 1.0, 2.0, 5.0\}$.

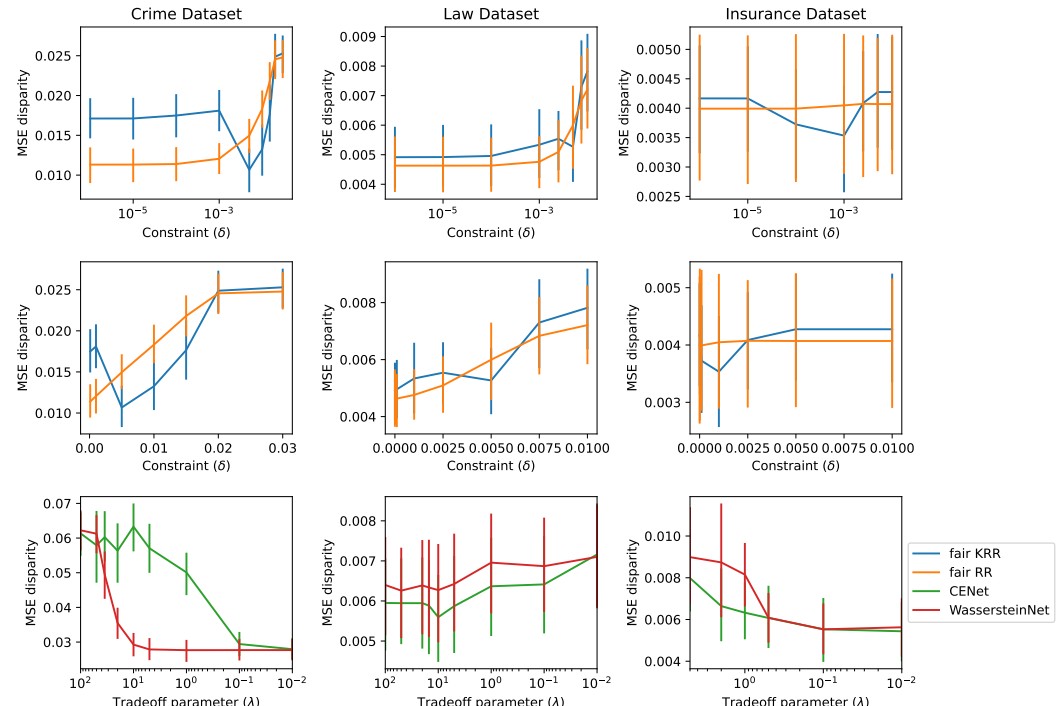

Figure 3: MSE disparity at various levels of fairness controlled by $\delta$ or $\lambda$. The model is supposed to be more "fair" with a smaller $\delta$ or larger $\lambda$. The standard error of 10 repetitions is represented by the error bars.

**Results and Analysis** For each repetition, we randomly split the data into 80% training and 20% testing set. We train the models on training data and assess their performance on testing set. The average metrics at each level of $\delta$ or $\lambda$ over 10 random splits are summarized in Figure 2 and Figure 3. Here are our observations: (1) Our proposed methods achieve the best trade-offs in all datasets. (2) Our methods effectively reduce the MSE disparity. Compare with fair RR, fair KRR works more effectively for relatively large $\delta$; when $\delta$ is small, the estimation of out-of-sample MSE disparity $\widehat{\Delta}(\hat{g}_\eta)$ exceeds the tuning limit $k \cdot \delta$, and a $\eta$ that minimizes the out-of-sample risk $\widehat{R}_\mathcal{T}(\hat{g}_\eta)$ will be selected by tuning rule. (3) Compared to our methods, the CENet method and WassersteinNet method can reduce the MSE disparity in some cases (such as in the Law Dataset), but overall, increasing $\lambda$ does not consistently reduce the MSE disparity as shown in Figure 3.

### B.3 NUMERICAL SIMULATIONS

In this section, we use simulations to illustrate the performance of the balancing weights proposed in Section 3 and our proposed optimization algorithms. In each simulation, we compare the balancing weights with inverse probability weights (IPW), proposed by Zhang & Long (2021). In particular, we compare with two IPWs, where the propensity score is obtained by fitting the selection indicator $S$ to fully observed covariates set $U$ with either a logistic regression (logistic) or a kernel support vector machine (kernel SVM) model with a Gaussian kernel. We also include the comparisons with the true IPW, even though it is usually unknown for real data. For balancing weights, a sobolev kernel is used for the optimization step.

We generate 500 replicates and each replicate consists of $n_1 = 2000$ and $n_0 = 2000$ samples. We showcase the results for four simulation settings and summarize the results in Table 1,2,3,4. For each experiment, we present the average MSE and average MSE disparity with their corresponding standard error. We highlight the smallest average MSE and MSE disparity at each level of fairness constraint $\delta$. The simulation results demonstrate that with the balancing weights adjustment, our proposed algorithm can consistently constrain the MSE disparity over the population distribution $p_\mathcal{T}$

Table 1: Mean and standard error of mse of weighted fair regression of case 1; all the values are multiplied by 100.

| $\delta$ | Weights | MSE | MSE disparity |
|---|---|---|---|
| | IPW (kernel SVM) | 675.587 (0.249) | 49.704 (1.559) |
| | IPW (logistic) | 669.682 (0.284) | 591.706 (2.666) |
| 0.0 | IPW (true) | 678.58 (0.273) | 42.168 (1.427) |
| | balancing weights | 677.949 (0.205) | **32.112 (1.039)** |
| | unweighted | **667.191 (0.234)** | 358.885 (1.852) |
| | IPW (kernel SVM) | 671.41 (0.229) | 85.749 (2.049) |
| | IPW (logistic) | 671.312 (0.294) | 624.165 (2.591) |
| 50.0 | IPW (true) | 673.814 (0.251) | 59.983 (1.744) |
| | balancing weights | 673.211 (0.189) | **53.755 (1.546)** |
| | unweighted | **667.763 (0.244)** | 341.495 (2.686) |
| | IPW (kernel SVM) | 668.423 (0.213) | 123.038 (2.115) |
| | IPW (logistic) | 672.855 (0.303) | 650.448 (2.533) |
| 90.0 | IPW (true) | 670.386 (0.234) | 91.26 (2.134) |
| | balancing weights | 669.804 (0.177) | **90.536 (1.736)** |
| | unweighted | **668.121 (0.252)** | 333.082 (3.163) |

even the fully observed training samples come from a distribution different from $p_{\mathcal{T}}$ under selection. Comparing to the IPW adjustments and even the true IPW, balancing weights adjustment can control the MSE disparity more effectively for most cases.

**Parametric regression:**

**Case 1:** Let $X_i = (X_{i1}, X_{i2}, X_{i3}, X_{i4})^T \in \mathbb{R}^4$ follow the Gaussian distribution such that $X_{ij} \sim N(0, \sigma_{A_i}^2)$, with $\sigma_0 = 1.5$ and $\sigma_1 = 1$. We let $U_i = (X_{i1}, X_{i2})^T$ denoting the covariates not subject to selection. The selection mechanism is based on the following model $S_i | A_i \sim$ Bernoulli (sigmoid($\pi_{A_i}(U_i)$)), where

$$\pi_0(U_i) = -2 + 0.1 \exp((U_{i1} - 1)/1.5) + 0.2((U_{i2} - 1)/0.8)^2,$$

$$\pi_1(U_i) = -1 + 2\sin(U_{i1} - 1)/1.2) - \log(|U_{i2}/5|).$$

Let $Y_i = X_i^T \boldsymbol{\beta} + (-4A_i + 1)X_{i2} + \epsilon_i$, for $\boldsymbol{\beta} = (1, -1, -1, 1)^T$, and $\epsilon_i \sim N(0, 1)$ independent of $X_i$ and $A_i$. In each repetition, we estimate the weights and apply Algorithm 1 with $\delta \in \{0, 0.5, 0.9\}$. We evaluate the performance on a sample of size $100,000$ from $p_{\mathcal{T}}$ and report the average and standard error of the metrics.

**Case 2:** Let $X_i = (X_{i1}, X_{i2}, X_{i3}, X_{i4})^T \in \mathbb{R}^4$ follow the Gaussian distribution such that $X_{ij} \sim N(1 - 2A_i, \sigma_{A_i}^2)$, with $\sigma_0 = 1.2$ and $\sigma_1 = 0.8$. We let $U_i = (X_{i1}, X_{i2})^T$ denoting the covariates not subject to selection. We adopt the same selection mechanism as above.

Let $Y_i = X_i^T \boldsymbol{\beta} + (-4A_i + 1)X_{i1} + \epsilon_i$, for $\boldsymbol{\beta} = (1, -1, -1, 1)^T$, and $\epsilon_i \sim N(0, 1)$ independent of $X_i$ and $A_i$. In each repetition, we estimate the weights and apply Algorithm 1 with $\delta \in \{0, 0.5, 0.9\}$. We evaluate the performance on a sample of size $100,000$ from $p_{\mathcal{T}}$ and report the average and standard error of the metrics.

**Nonparametric regression**:

**Case 3** Let $\widetilde{X}_i = (\widetilde{X}_{i1}, \widetilde{X}_{i2}, \widetilde{X}_{i3}, \widetilde{X}_{i4})^T \in \mathbb{R}^4$ follow the gaussian distribution such that $\widetilde{X}_{ij} \sim N(1 - 2A_i, 1)$.

The selection mechanism is based on the following model $S_i | A_i \sim$ Bernoulli(sigmoid($\pi_{A_i}(\widetilde{X}_i)$)), where

$$\pi_0(\widetilde{X}_i) = 1 - 0.8\widetilde{X}_{i1} - 0.2\widetilde{X}_{i2} - 0.25\widetilde{X}_{i3} - 0.1\widetilde{X}_{i4},$$

Table 2: Mean and standard error of mse of weighted fair regression of case 2; all the values are multiplied by 100.

| $\delta$ | Weights | MSE | MSE disparity |
|---|---|---|---|
| | IPW (kernel SVM) | **513.742 (0.389)** | 131.985 (3.54) |
| | IPW (logistic) | 536.328 (1.461) | 70.601 (3.052) |
| 0.0 | IPW (true) | 515.693 (0.489) | 51.549 (2.247) |
| | balancing weights | 514.181 (0.43) | **40.123 (1.649)** |
| | unweighted | 570.96 (1.192) | 362.96 (4.812) |
| | IPW (kernel SVM) | 512.369 (0.391) | 182.804 (3.707) |
| | IPW (logistic) | 533.645 (1.423) | 82.148 (3.446) |
| 50.0 | IPW (true) | 510.965 (0.427) | 63.402 (2.759) |
| | balancing weights | **509.475 (0.384)** | **53.168 (2.19)** |
| | unweighted | 574.145 (1.256) | 402.068 (5.372) |
| | IPW (kernel SVM) | 512.3 (0.411) | 224.591 (3.801) |
| | IPW (logistic) | 531.923 (1.393) | 97.571 (3.899) |
| 90.0 | IPW (true) | 508.107 (0.381) | 92.426 (3.349) |
| | balancing weights | **506.602 (0.349)** | **85.324 (2.711)** |
| | unweighted | 576.13 (1.309) | 421.302 (5.835) |

Table 3: Mean and standard error of mse of weighted fair regression of case 3; all the values are multiplied by 100.

| $\delta$ | Weights | MSE | MSE disparity |
|---|---|---|---|
| | IPW (kernel SVM) | 675.022 (89.549) | 285.889 (54.493) |
| | IPW (logistic) | 326.999 (27.337) | **93.419 (13.771)** |
| 10.0 | IPW (true) | 398.192 (52.417) | 155.848 (33.64) |
| | balancing weights | 321.863 (9.744) | 94.404 (5.168) |
| | unweighted | **321.191 (18.824)** | 124.904 (12.843) |
| | IPW (kernel SVM) | 611.08 (95.804) | 288.263 (61.81) |
| | IPW (logistic) | 314.115 (33.503) | 99.25 (15.653) |
| 40.0 | IPW (true) | 359.299 (56.734) | 135.306 (32.964) |
| | balancing weights | 279.165 (13.448) | **81.353 (4.962)** |
| | unweighted | **260.925 (12.145)** | 89.738 (8.583) |
| | IPW (kernel SVM) | 619.388 (95.285) | 295.261 (51.575) |
| | IPW (logistic) | 302.156 (29.485) | 117.268 (13.972) |
| 80.0 | IPW (true) | 384.351 (62.054) | 162.384 (33.715) |
| | balancing weights | 246.321 (6.007) | **82.065 (3.797)** |
| | unweighted | **225.814 (4.298)** | 90.585 (4.218) |

$$\pi_1(\widetilde{X}_i) = 0.5 + 0.5\widetilde{X}_{i1} - 2\widetilde{X}_{i2} - 0.2\widetilde{X}_{i3} - 0.1\widetilde{X}_{i4}.$$

The observed covariates $X_{i1} = \exp(\widetilde{X}_{i1}/2)$, $X_{i2} = \widetilde{X}_{i2}/(1 + \exp(\widetilde{X}_{i1})) + 10$, $X_{i3} = (\widetilde{X}_{i3}/25 + 0.6)^3$, $X_{i4} = (\widetilde{X}_{i2} + \widetilde{X}_{i4} + 20)^2$. We let $U_i = (X_{i1}, X_{i2}, X_{i3}, X_{i4})^T$ denoting the covariates not subject to selection. Let $Y_i = 27.4\widetilde{X}_{i1} + 13.7\widetilde{X}_{i2} + 13.7\widetilde{X}_{i3} + 13.7\widetilde{X}_{i4} + (-2A_i + 0.5)\widetilde{X}_{i1}(\widetilde{X}_{i3} + \widetilde{X}_{i4})/2 + \epsilon_i$ with $\epsilon_i \sim N(0, 1)$ independent of $\widetilde{X}_i$ and $A_i$.

For each repetition, we estimate the weights and apply Algorithm 1 with $\delta \in \{0.1, 0.4, 0.8\}$. We select the penalizing hyper-parameter $\eta$ from 20 evenly spaced numbers on a log scale between $10^{-8}$ and $10^{-1}$. We evaluate the performance on a sample of size 2000 from $p_{\mathcal{T}}$ and report the average and standard error of the metrics.

Table 4: Mean and standard error of mse of weighted fair regression of case 4; all the values are multiplied by 100.

| $\delta$ | Weights | MSE | MSE disparity |
|---|---|---|---|
| | IPW (kernel SVM) | 988.292 (146.981) | 212.571 (32.191) |
| | IPW (logistic) | 986.541 (109.23) | 239.749 (29.474) |
| 10.0 | IPW (true) | 1148.411 (157.506) | 214.139 (34.216) |
| | balancing weights | **655.782 (85.631)** | **132.995 (18.233)** |
| | unweighted | 1770.393 (128.721) | 622.058 (49.221) |
| | IPW (kernel SVM) | 1095.237 (157.847) | 272.822 (38.877) |
| | IPW (logistic) | 891.51 (104.671) | 223.875 (25.495) |
| 40.0 | IPW (true) | 994.546 (144.07) | 218.461 (32.34) |
| | balancing weights | **527.704 (71.915)** | **138.869 (15.876)** |
| | unweighted | 962.436 (98.601) | 349.391 (37.017) |
| | IPW (kernel SVM) | 507.14 (97.388) | 184.18 (24.907) |
| | IPW (logistic) | 399.367 (58.096) | 165.881 (16.936) |
| 80.0 | IPW (true) | 591.531 (108.822) | 166.445 (20.541) |
| | balancing weights | 424.582 (59.104) | **139.38 (9.321)** |
| | unweighted | **301.806 (39.492)** | 158.556 (16.41) |

**Case 4** Let $\widetilde{X}_i = (\widetilde{X}_{i1}, \widetilde{X}_{i2}, \widetilde{X}_{i3}, \widetilde{X}_{i4})^T \in \mathbb{R}^4$ follow the gaussian distribution such that $\widetilde{X}_{ij} \sim N(1 - 2A_i, \sigma_{A_i}^2)$, with $\sigma_0^2 = 0.8$ and $\sigma_1^2 = 1$.

The selection mechanism is based on the following model $S_i | A_i \sim \text{Bernoulli}(\text{sigmoid}(\pi_{A_i}(\widetilde{X}_i)))$, where

$$\pi_0(\widetilde{X}_i) = 1 - 0.1\widetilde{X}_{i1} - 0.1\widetilde{X}_{i2} - 0.2\widetilde{X}_{i3} - 0.2\widetilde{X}_{i4},$$

$$\pi_1(\widetilde{X}_i) = 0.5 + 0.5\widetilde{X}_{i1} - 2\widetilde{X}_{i2} - 0.2\widetilde{X}_{i3} - 0.1\widetilde{X}_{i4}.$$

The observed covariates $X_{i1} = \exp(\widetilde{X}_{i1}/2)$, $X_{i2} = \widetilde{X}_{i2}/(1 + \exp(\widetilde{X}_{i1})) + 10$, $X_{i3} = (\widetilde{X}_{i3}/25 + 0.6)^3$, $X_{i4} = (\widetilde{X}_{i2} + \widetilde{X}_{i4} + 20)^2$. We let $U_i = (X_{i1}, X_{i2}, X_{i3}, X_{i4})^T$ denoting the covariates not subject to selection. Let $Y_i = 27.4\widetilde{X}_{i1} + 13.7\widetilde{X}_{i2} + 13.7\widetilde{X}_{i3} + 13.7\widetilde{X}_{i4} + (-2A_i + 0.5)\widetilde{X}_{i1}(\widetilde{X}_{i3} + \widetilde{X}_{i4})/2 + \epsilon_i$ with $\epsilon_i \sim N(0, 1)$ independent of $\widetilde{X}_i$ and $A_i$.

For each repetition, we estimate the weights and apply Algorithm 1 with $\delta \in \{0.1, 0.4, 0.8\}$. We select the penalizing hyper-parameter $\eta$ from 20 evenly spaced numbers on a log scale between $10^{-8}$ and $10^{-1}$. We evaluate the performance on a sample of size 2000 from $p_{\mathcal{T}}$ and report the average and standard error of the metrics.

### B.4 COMPARISON WITH DISCIPLINED CONVEX-CONCAVE PROGRAMMING (DCCP)

In this section, we follow the same setting as in case 3 and compare the performance of our optimization method and DCCP. We fix $\eta = 10^{-6}$ and do not apply any weights adjustment to both methods. The MOSEK optimizer is employed to solve DCCP with default parameter configurations. For all repetitions, both DCCP and our method yield feasible solutions, with the MSE disparity bounded by $\delta + 10^{-6}$. We compare the average time and maximal time needed to finish training for one repetition. We also include frequency of each method achieving a smaller objective value. The summarized results from 500 random repetitions are presented in Table 5. Each repetition is completed with one node with 2 CPUs, which has 8 cores and 16 threads. Each compute node has 64 GB of RAM. The results show that our method is significantly more efficient than DCCP, and consistently achieving a smaller objective value.

Table 5: Comparison between lagrangian and dccp optimization.

| $\delta$ | Optimization | Average time (s) | Maximal time (s) | Smaller objective value frequency |
|---|---|---|---|---|
| 0.1 | Lagrangian | 0.577 | 0.723 | 500 |
|  | DCCP | 71.916 | 109.843 | 0 |
| 0.4 | Lagrangian | 0.577 | 0.738 | 500 |
|  | DCCP | 66.674 | 107.829 | 0 |
| 0.8 | Lagrangian | 0.575 | 0.684 | 500 |
|  | DCCP | 56.239 | 109.155 | 0 |

## C  NEWTON'S METHOD IMPLEMENTATION

In this section, we present the implementation details of the Newton's method that will be used to optimize the objective function in the dual problem (20).

---

**Algorithm 2** Optimization procedure to solve $\lambda^* = \text{argmax } h_C(\lambda)$.

---

1: $\lambda_0 = 0$
2: **while** epoch < maxepoch; epoch ++ **do**
3:      $\Delta\lambda = \frac{\nabla h_C(\lambda_i)}{\nabla^2 h_C(\lambda_i)}$
4:      $\nabla h_C(\lambda_i) = C + 2\boldsymbol{Y}^T \boldsymbol{D} \boldsymbol{L} \boldsymbol{\beta}(\lambda_i) - \boldsymbol{\beta}(\lambda_i)^T (\boldsymbol{L}^T \boldsymbol{D} \boldsymbol{L}) \boldsymbol{\beta}(\lambda_i),$
5: $\nabla^2 h_C(\lambda_i) = -2(\boldsymbol{\beta}(\lambda_i) \boldsymbol{L}^T \boldsymbol{D} \boldsymbol{L} - \boldsymbol{L}^T \boldsymbol{D} \boldsymbol{Y})^T (\boldsymbol{L}^T \boldsymbol{W} \boldsymbol{L} - \lambda_i \boldsymbol{L}^T \boldsymbol{D} \boldsymbol{L} + \eta \mathbf{I})^{-1} (\boldsymbol{\beta}(\lambda_i) \boldsymbol{L}^T \boldsymbol{D} \boldsymbol{L} - \boldsymbol{L}^T \boldsymbol{D} \boldsymbol{Y})$
6:      **if** $\Delta\lambda > 0$ **then**
7:          $\Delta\lambda = \min(\Delta\lambda, 0.95(\bar{\lambda} - \lambda_i))$
8:      **else**
9:          $\Delta\lambda = \max(\Delta\lambda, 0.95(\underline{\lambda} - \lambda_i))$
10:      **end if**
11:      $\lambda_{i+1} = \lambda_i + \Delta\lambda; \boldsymbol{\beta}_{i+1} = \boldsymbol{\beta}(\lambda_{i+1})$
12:      **if**

$$\max\left\{\frac{|\nabla h_C(\lambda_i)|}{|S|+1}, \frac{|f_0(\boldsymbol{\beta}_{i+1}) - h(\lambda_{i+1})|}{|f_0(\boldsymbol{\beta}_{i+1})|+1}, \left(\frac{||(\boldsymbol{L}^T(\boldsymbol{W} - \lambda_{i+1}\boldsymbol{D})\boldsymbol{L})\boldsymbol{\beta}_{i+1} - (\boldsymbol{L}^T(\boldsymbol{W} - \lambda_{i+1}\boldsymbol{D})\boldsymbol{Y})||}{||\boldsymbol{L}^T(\boldsymbol{W} - \lambda_{i+1}\boldsymbol{D})\boldsymbol{L}|| + ||\boldsymbol{L}^T(\boldsymbol{W} - \lambda_{i+1}\boldsymbol{D})\boldsymbol{Y}|| + 1}\right)^2\right\} \leq tol$$

     **then**
13:          Break.
14:      **end if**
15: **end while**
16: **Return** $\lambda_{i+1}$

---

