# OpenReview forum: "Weighted Fair Regression under Selection Bias"
_ICLR.cc/2025/Conference — ICLR 2025 Conference Withdrawn Submission_

### Official Review · Reviewer_Lejf · 2024-10-21

**Soundness:** 4
**Presentation:** 2
**Contribution:** 3
**Rating:** 5
**Confidence:** 3

**Summary:**

This paper presents a novel approach for addressing unfairness in regression due to selection bias. For example, imagine some dataset, and then imagine a filter is applied over that dataset before modeling: how can we ensure that a regression model trained on the filtered data is fair with respect to the data *before* the filter was applied? The authors propose a method that uses weighting adjustments.

To the best of my understanding, if, for example, many individuals from a protected group were filtered out of the dataset, one might expect larger weights for the individuals from that group that *did* make into it the filtered dataset.

**Strengths:**

Unfortunately (as indicated by my confidence score of 3) theory is not my expertise, which makes up a majority of the paper. In this area, I have to defer to other reviewers and the AC chair.

However, from what I can understand, the authors have presented a strong theoretical foundation for their work, and after reviewing Du et al. 2022, it appears the authors have made meaningful contributions over the state-of-the-art.  Further, while I cannot validate them, claims like "despite non-convexity we derive an efficient algorithm to obtain a global optimal solution" (abstract), and "the results demonstrate the proposed algorithm consistently achieves a smaller objective value, and is significantly more efficient... reducing both the average and maximum computation time to less than 1% of DCCP optimization" (lines 257-259) are noteworthy.

From a fairness perspective (which is my area of expertise), the setting makes sense, and I am convinced by the motivating example (lines 123-130) (although I would make the adjustment that the system evaluates the chances that a candidate *gets an interview* rather than their interview performance, as the former is observed much more often in the wild [1]).

[1] https://eightfold.ai/

**Weaknesses:**

- Choice of fairness metric: the authors are right that fairness in regression isn't nearly as well studied as in classification... but that's because the strong choice of a fairness metric often implies a downstream classification task [1]. For example, I could imagine that in Example 1, the true, consequential underlying fairness issue is that the recruitment algorithm would result in recruitment biased against individuals from a protected class (meaning that disparities in the regression score are upstream of a classification decision for who actually gets recruited vs. who doesn't). But, all that aside, and accepting that there are settings where evaluating fairness on a continuous score makes sense (ex. an algorithm that is used to calculate property tax), why use MSE as opposed to any other continuous measure of accuracy (beyond the fact that other authors have used it)? Could the authors provide justification for this metric, or clarify whether or not this method can generalize to other continuous fairness metrics? This would give readers a better understanding of the scope and applicability of the proposed approach.

- Motivation issues: Lines 33-42 motivate the issue of selection bias very well and highlight why they need to be solved. However, I felt that Assumption 1 undermined some of that motivation... as noted on lines 40-42, nearly any data $Z$ we collect from the world will suffer from selection bias-- so there would exist no such vector $U$ that is not subject to selection bias. Could the authors address how Assumption 1 can be reconciled with real-world data collection challenges, or discuss potential approaches for estimating or simulating an unbiased $U$ vector when truly unbiased data may not be available.

- Regarding the claim "the results demonstrate the proposed algorithm consistently achieves a smaller objective value, and is significantly more efficient... reducing both the average and maximum computation time to less than 1% of DCCP optimization" (lines 257-259): this feels like an important claim, but there is not any evidence for it in the main body of the paper. Can the authors provide evidence (e.g. experimental results) supporting this claim? Otherwise the authors might consider removing this claim.

- Experimental evaluation: I'm assuming that the experimental evaluation was probably not the focus of the authors, given the strong theoretical focus of the paper, but it still left me with several questions. First, I found the results for crime data (2) in Figure 1 to be a bit dubious: there was such a large drop in the test overall MSE. If these are accurate (which they very well may be), can the authors provide intuition as to why their method performed so much better than the baseline? (This is particularly hard for me to reconcile because the selling point of the method was that it didn't require as strong assumptions as the baseline -- it isn't obvious to me why changes in assumptions => better performance...)

- Experimental evaluation continued: What I'm really left wondering about all the experiments, is if the performance improvements over Du et al. are due to improvements by the proposed approach, or just a byproduct of better fine-tuning by the authors. Could the authors clarify how they controlled for implementation differences and hyperparameter tuning when comparing to Du et al., to ensure a fair comparison? Further, Du et al. also use a third dataset (COMPAS) -- could the authors explain why that was not included?

- Beyond the theoretical contributions: again, while the theoretical contributions appear strong, I'm left with a bit of a "so what?" There is no mention of an implementation, guidance to practitioners, when this would be used, strong use cases, etc. Fairness is an inherently applied field that ultimately seeks to help real-world individuals from marginalized and vulnerable populations. Perhaps the authors could include a section discussing potential real-world applications, implementation guidelines for practitioners, and specific use cases where their method could have meaningful impact on fairness issues affecting marginalized populations.

Note to the authors: I am very willing to increase my score based on your responses and based on other reviewer/AC chair comments. My core challenge here is reconciling what appears to be a strong theoretical contribution (for which *I am not* the right reviewer), and then weaknesses in the motivation (fairness framing) and experimentation (for which *I am* the right reviewer)...

[1] https://www.datasciencepublicpolicy.org/our-work/tools-guides/aequitas/

**Questions:**

Combined with above section.

---

### Official Review · Reviewer_Q8pa · 2024-10-28

**Soundness:** 3
**Presentation:** 3
**Contribution:** 1
**Rating:** 3
**Confidence:** 3

**Summary:**

The paper considers the problem of fair linear regression where there is a selection bias in the training data. Their motivating example is a hiring setting where the covariates are information about a candidate, the protected attribute is gender, and the outcome is the quality of the interview. Because of biases, it may be more likely for men to get to the interview so the training dataset is biased against women and does does not reflect the true distribution.

The first part of this work considers how to reweight the observations in the empirical loss function to approximate the empirical loss on the true distribution. They add and subtract various terms to this weighted empirical loss so that the weighted empirical loss is the true empirical loss plus the difference of several terms. They argue that two of the differences in terms will be small for a large number of samples and then they choose the weights to minimize an upper bound on the third term.

The second part of this work considers how to solve the regression problem once we have these weights. They formulate a constrained linear regression problem with a normalization penalty. They then turn this into a Lagrangian form and, under several assumptions, derive the optimal solution. This formulation has been considered before but they solve the dual version rather than the primal version. The advantage is that, under several assumptions, they can approximate the leave-one-out-error by (essentially) subtracting the predicted outcome for the left out observation.

They test their method on two datasets (law and crime). The results indicate that all perform quite similarly and theirs is slightly better sometimes.

**Strengths:**

* The paper considers an interesting problem

* The paper uses many techniques from prior work, demonstrating familiarity with relevant literature

**Weaknesses:**

My general feeling after reading this work is that they put in lots of notation and algebra to try to impress the reader. Most of the steps they take are a) from prior work or b) seem arbitrary without proper justification.

* In terms of the reweighting section, they make arbitrary choices about what terms to add to their estimator. They then wave away several resulting terms with an asymptotic argument that is not satisfying. For the term they do optimize, they optimize an upper bound. When I looked at their method for optimizing it in the appendix, they use a further upper bound in terms of a (arbitrary as far as I could tell) optimization formulation.

* Once they choose the weights, the rest of the problem is independent of the weights and has already been studied before in Pong & Wolkowicz (2014). The current paper solves the dual problem instead of the primal which they claim allows them to compute leave one out error more efficiently. However, they make lots of assumptions to argue that the leave one out error is approximated. If you're willing to make lots of assumptions and consider the asymptotic setting (as they are), then you might as well argue that leave one out error is approximated by the full error which, morally, seems like what they're doing.

* Their experimental setting is very unpersuasive. They only consider two datasets. They then report the results in a very misleading way: making the range of the x and y axes very small (e.g. between .0546 and .0556) so that the marginal improvement of their algorithm seems bigger than it is. Reading the three figures, the SVM inverse propensity weighting approach seems about as good as theirs and, because of the small range, the algorithms all perform essentially the same. To be convinced by the experiments, I'd want to see many more datasets and performance under various hyperparameter settings.

**Questions:**

* Does Assumption 1 mean that there is an observation U = X_1, Y_1 so that, conditioning on this covariate and outcome, the training data approximates the loss on the true distribution?

* Why do you use the optimization formulation of Wong & Chan (2014) when choosing the weights in Appendix A.1?

* What is the time complexity of Algorithm 1?

---

### Official Review · Reviewer_mzG6 · 2024-10-28

**Soundness:** 3
**Presentation:** 2
**Contribution:** 2
**Rating:** 5
**Confidence:** 3

**Summary:**

The paper studies regression with missing data with the fairness constraint that the loss for each group has to be also roughly equal. Under the assumption that conditioned on the observed variables, the expected value of the loss is the same over the target and observed population, the paper proposes a reweighting scheme to enforce fairness and proposes an algorithm to find a fair regressor.

**Strengths:**

-- The paper studies an important and interesting problem.

**Weaknesses:**

-- The paper is hard to read. Saying in words what equations/assumptions etc mean can improve the readability of the paper.

-- Many technical details are relegated to the appendix. For example. the results from weight adjustments are relegated to Appendix A.1. Without these details it is hard to assess the technical novelty. In particular, a quick scan of the appendix shows that many of the results are derived by relying on prior work (e.g. Wong & Chen 2018). I think the details of these results should be moved to the body of the paper.

-- The empirical analysis does not seem to be convincing. The small range in the axes shows that there is not that much variability between different techniques (see questions below).

**Questions:**

-- Assumption 1 still feels strong to me. Why should this assumption hold in your screening example?

-- Can the authors explain clearly how their work is technically different from some prior cited work? What techniques are used in prior work to address the condition mentioned in line 138?

-- What makes equal loss fairness important under the paper's setting? Usually, equal loss is important when there is a different amount of data available for each group. Here I think there is heteroscedasticity in the noise but enough sample is given from each population. Is this correct? If so, can the authors provide an example where the notion makes sense?

-- In fairness literature, usually, the trade-off between fairness violation and error is studied. This can be achieved by varying the delta in Equation 1. What do these trade-offs look like for your approach and other approaches? Does varying delta allow us to better differentiate between the performance of different algorithms? Alternatively, experimenting on different datasets with more variability between the different approaches would be a better way to showcase the effectiveness of your method.

---

### Official Review · Reviewer_N1VK · 2024-11-01

**Soundness:** 3
**Presentation:** 3
**Contribution:** 3
**Rating:** 6
**Confidence:** 3

**Summary:**

The authors study the problem of fairness regression when there is selection bias in training data. To address the unfairness which may arise from such bias the authors propose a scheme to reweight the fairness constraint such that training unfairness and testing unfairness are more closely aligned.

**Strengths:**

- The combination of fair regression and selection bias is an important problem.

- The authors propose a novel and intuitive method for attaining fair regressors in the case of selection bias.

- The authors provide several theoretical results as well as useful derivations (such as a dual formulation and conditions under which strong duality holds), which help contextualize their approach.

- The paper is mostly well-written. The authors provide useful motivation or interpretation for many of their assumptions and results. For example, the interpretations of the conditions in Assumption 2 are helpful. Of course, readers could work out this condition out on their own, but it is always helpful to hear interpretations from those who have been thinking deeply about the given problem.

- The authors motivate their Lagrangian-based approach (Algorithm 1) through increased efficiency. I believe the results in Table 5 of the appendix are also support this notion (if so, it may be a good idea to reference this table somewhere in the main body).

**Weaknesses:**

1) The paper overclaims their contributions at times. For example, the abstract states that
> “This work pioneers the integration of weighting adjustments into the fair regression problem.”

However, many works have used weighting, even dynamic weighting, in the context of fair regression (the authors cite such works). Perhaps one of the earliest examples is that of [1], which uses cost-sensitive learning with dynamically changing weights to learn fair regressors.

2) My biggest concern with this paper is the limited empirical results. The authors show a single set of results in the main body, which covers only a single example of selection bias (shown on lines 479 for Crime and 501 for Law). Given that fair regression, and selection bias, are both highly practical problems, I would have expected to see a deeper empirical analysis. In particular, I would have liked to see results indicating

    a) how each method (the authors’ and the baselines) functions as selection bias becomes more aggressive,

    b) how well each method performs when there is no selection bias (i.e., answering the question of whether the authors' method can create fair regressors even when unfairness does not stem from selection bias), and

    c) results for different sensitive features.

3) The authors' method appears to be applicable only to case of two groups.



### References
[1] Agarwal, Alekh, Miroslav Dudík, and Zhiwei Steven Wu. "Fair regression: Quantitative definitions and reduction-based algorithms." International Conference on Machine Learning. PMLR, 2019.

**Questions:**

Please address my comments in the Weaknesses section.

---

### Note · Authors · 2024-11-20

I have read and agree with the venue's withdrawal policy on behalf of myself and my co-authors.